# OUTPACE long duration stations: physical variability, context of biogeochemical sampling, and evaluation of sampling strategy

Alain de Verneil[1], Louise Rousselet[1], Andrea M. Doglioli[1], Anne A. Petrenko[1], Christophe Maes[2], Pascale Bouruet-Aubertot[3], and Thierry Moutin[1]

[1]Aix-Marseille University, MIO, CNRS/INSU, Marseille, France
[2]Univ. Brest, Ifremer, CNRS, IRD, Laboratoire d'Océanographie Physique et Spatiale (LOPS), IUEM, F-29280, Brest, France
[3]LOCEAN, UMR7167, Université Pierre et Marie Curie, 75252 Paris, France

*Correspondence to:* Alain de Verneil (alain.de-verneil@mio.osupytheas.fr)

**Abstract.** Research cruises to quantify biogeochemical fluxes in the ocean require taking measurements at stations lasting at least several days. A popular experimental design is the quasi-Lagrangian drifter, often mounted with in situ incubations that follow the flow of water over time. After initial drifter deployment, the ship tracks the drifter for continuing measurements that are supposed to represent the same water environment. An outstanding question is how to best determine whether this is true.
During the Oligotrophy to UlTra-oligotrophy PACific Experiment (OUTPACE) cruise, from 18 February to 3 April 2015 in the western tropical South Pacific, three separate stations of long duration (five days) over the upper 500 m were conducted in this quasi-Lagrangian sampling scheme. Here we present physical data to provide context for these three stations and to assess whether the sampling strategy worked, ie that indeed a single body of water was sampled. After analyzing tracer variability and local water circulation at each station, we identify water layers and times where the drifter risks encountering another
body of water. While almost no realization of this sampling scheme will be truly Lagrangian due to the presence of vertical shear, the depth-resolved observations during the three stations show most layers sampled sufficiently homogeneous physical environments during OUTPACE. By directly addressing the concerns raised by these quasi-Lagrangian sampling platforms, a protocol of best practices can begin to be formulated so that future research campaigns include the complementary datasets and analyses presented here to verify the appropriate use of the drifter platform.

## 1  Introduction

Biogeochemical cycles dictate the global distribution and fluxes of the chemical elements. Quantifying the mechanisms that mediate the various forms key elements take in these cycles, especially in the midst of ongoing climate change in the ocean, is vital to understanding the future evolution of the Earth system (Falkowski et al., 2000; Davidson and Janssens, 2006; Gruber and Galloway, 2008). Considering the wide diversity of environments where biogeochemical processes take place, it is not
surprising that each sub-discipline has its own challenges with regards to collecting and processing samples. The sampling protocols put in place thus need to ensure the mechanisms of interest are isolated and put into their proper context.

In the world's surface oceans, a dominant difficulty is the medium itself: water. Sampling in a fluid that is always liable to move normally requires that one of two approaches be taken. In the first approach, a geographic location is chosen and

then repeatedly sampled. This produces an Eulerian perspective, and this methodology is employed by definition at permanent mooring platforms. Set geographic locations are also often used to define time series or recurrent sampling locations, for example stations ALOHA, BATS, CalCOFI, DYFAMED, and PAPA (Karl and Lukas, 1996; Schroeder and Stommel, 1969; Steinberg et al., 2001; Bograd et al., 2003; Marty et al., 2002; Freeland, 2007). These sites can be combined into world-wide networks and initiatives such as OceanSITES (Send et al., 2010). While this strategy makes no attempt to actually follow a given water parcel, if currents are relatively weak during a single field campaign then the variability due to advection can be ignored. Unfortunately, the spatio-temporal scales of shipboard station sampling (in time, days to weeks; in space, 1-100 km) overlap with a multitude of physical phenomena ubiquitously found in the ocean, ranging from internal waves, submesoscale turbulence, up to mesoscale eddies (d'Ovidio et al., 2015). All of these motions can easily transport water such that instantaneously observed temperature, salinity, and by extension the organisms and chemical environments mediating biogeochemical processes, are markedly different from some mean value or state.

One way to rectify physical displacements is the second sampling approach, namely to follow the water during ongoing experiments. This approach creates a Lagrangian point of view. A common implementation of this strategy is with quasi-Lagrangian drifting moorings (Landry et al., 2009; Moutin et al., 2012). These drifters are structured so that a vertical line with sampling devices (e.g. incubation bottles and/or sediment traps) drifts along with the flow. This approach has been in routine use for decades across the globe; some examples of French campaigns known to the authors include the OLIPAC (1994), PROSOPE (1999), BIOSOPE (2004), and BOUM (2008) experiments (all data and metadata accessible from LEFE-CYBER).

Naturally, the question arises whether the trajectory undertaken by the drifting mooring in the quasi-Lagrangian approach accurately represents the water movements at each of the sampling sites. If the drifter is successful in following the water, then indeed a single biogeochemical setting will have been sampled; if not successful, then the risk grows that a different environment has been brought in via advection. Previous efforts by physicists to make floats Lagrangian show the effort needed to make an instrument neutrally buoyant, and these floats have been instrumental in demonstrating complicated flow regimes (D'Asaro et al., 2011). In contrast, the quasi-Lagrangian platform, with a variable distribution of incubation bottles, will necessarily fail to be Lagrangian in finite time outside of a barotropic flow where currents are the same throughout the water column. As a result, ensuring the success of this strategy requires taking into account how different currents potentially shorten the timespan of validity. In fast-moving flows with strong vertical shear and possible vertical motions, such as zones of enhanced mesoscale turbulence near fronts and filaments, the drifter will not be Lagrangian long. Alternatively, if a drifter is trapped inside a coherent eddy, it can follow a similar water mass for a long time over great distances. Therefore, before deployment the selection of sampling sites needs to be carefully considered. Unless the focus of study, fronts and filaments need to be avoided because shearing will quickly separate water parcels at different depths in the direction of the structure's alignment; finding signs of their presence has become more feasible with satellite data. An eddy can be targeted because of its coherence, and there are ways to confirm that sampling is indeed inside of it (Moutin and Prieur, 2012). In other words, if a physical structure is targeted or identified, its particular nature supercedes other considerations. These structures are not necessarily representative of the world Ocean, and so many biogeochemical measurements will be taken elsewhere. For the

campaigns where sites are far (possibly by design) from obvious, organized mesoscale structure, there is still a need to conduct an independent, post-cruise validation of the drifter's success, which is the focus of the present study.

Before proceeding into this study's description of our methodology, a few remarks are needed regarding its applicability. We already mentioned that we will consider regions away from strong, organized mesoscale structure. Additionally, the method relies upon independent physical, not biogeochemical, measurements to indicate a change of water mass due to the drifter not being Lagrangian. This approach does not detect the existence of biogeochemical gradients in water that might exist on smaller scales, so application of our method requires the user to apply contextual knowledge of their sampling region and keep this possibility in mind. For this study, a regional biogeochemical gradient was expected (Moutin et al., 2017) and rationales for this method's application will be provided.

The Oligotrophy to UlTra-oligotrophy PACific Experiment (OUTPACE) cruise provided an opportunity to assess the success of the quasi-Lagrangian sampling strategy. Conducted from 18 February to 3 April 2015 in the western tropical South Pacific (WTSP), one of the goals of OUTPACE was to assess the regional contribution of nitrogen fixation as a biogeochemical process to the biological carbon pump (Moutin et al., 2017). During the cruise, three long duration (LD) stations employed the quasi-Lagrangian strategy. In the subsequent discourse regarding these stations, we proceed as follows. Sect. 2 describes how the drifting mooring was deployed, our methodological strategy, how concurrent data were collected, and the analyses undertaken to answer our central question of whether we sampled a single environment. We then present the data and results in Sect. 3, followed by a discussion in Sect. 4. The paper finishes in Sect. 5 with a summary of our recommendations regarding future implementations of this sampling strategy.

## 2   Materials and Methods

In this section, we begin by describing the general manner in which the three LD stations were conducted during the OUTPACE cruise. Following an outline of the methodological strategy, we present the different data sources and their processing. Additionally, we describe in detail the analyses needed to answer our central question regarding sampling in a coherent environment.

### 2.1   Sampling strategy

The OUTPACE cruise occurred aboard the RV *L'Atalante* from 18 February to 3 April in late austral summer, starting in New Caledonia and finishing in Tahiti, traveling over 4000 km. Stations were conducted in a mostly zonal transect traveling west to east, with the ship track averaging near 19° S. The three LD stations, entitled LDA, LDB, and LDC, and lasting 5 days each, were designed to resolve a regional zonal gradient in oligotrophy, the existence of which is reflected in the surface chlorophyll-*a* (chl-*a*) data (Fig. 1a). As described in the introductory article of this special issue (Moutin et al., 2017), site selection for the LD stations involved identifying physical structures by use of the SPASSO software package (http://www.mio.univ-amu.fr/SPASSO/) using near-real-time satellite imagery, altimetry, and Lagrangian diagnostics (Doglioli et al., 2013; d'Ovidio et al., 2015; Petrenko et al., 2017).

Before starting each LD station, surface velocity program (SVP; Lumpkin and Pazos, 2007) drifters were deployed adjacent to the site. The number of drifters deployed are summarized in Table 1, and their mean initial positions were 1.1, 1.6, and 0.9 km away from the first CTD of station LDA, LDB, and LDC, respectively. At the start of each station, two quasi-Lagrangian drifting moorings were deployed during the OUTPACE LD stations with surface floats. The first drifting mooring, hereafter referred to as the SedTrap Drifter, had a 'holey sock' attached at 15 m depth. It was followed actively by the ship and is the emphasis of this study. It had three sediment traps (Technicap PPS5/4) fixed at 150, 250, and 500 m depth, along with onboard conductivity-temperature-depth (CTD) sensors and current meters, described below in Sects. 2.4.1 and 2.6, respectively. The SedTrap Drifter was deployed at the beginning of each station and was left in the water until the station's completion. The second drifting mooring, referred to as the Production Line, housed in situ incubation platforms for measuring primary production, nitrogen fixation, oxygen, and other biogeochemical measurements (see Moutin and Bonnet, 2015, for more documentation). The Production Line was redeployed on a daily basis close to the SedTrap Drifter. While no telemetry exists for the Production Line, the CTD casts from which incubation water was drawn ranged from 300 m to 5.7 km from the SedTrap drifter. After 5 days, the SedTrap Drifter was recovered, and the LD station completed. Occasions when the exact implementation of this general strategy was not realized will be mentioned in following sections for the relevant measurements. A summary of time duration for each data source can be found in Table 1.

Between the LD stations, 15 short duration (SD) stations lasting approximately 8 h each were interspersed along the ship's trajectory in roughly equidistant sections (Fig. 1b). Among the measurements made, CTD casts from SD stations will figure into the validation process in this study. Most casts (both LD and SD) were at least 200 m, with at least one 2000 m cast for all stations. These casts were conducted with the same CTD rosette platform described more fully below in Sect. 2.3.1.

Throughout the cruise, surface conductivity-temperature measurements from the themosalinograph (TSG) and currents from shipboard acoustic Doppler current profilers (SADCP) were collected. Their processing is described in Sects. 2.4.1 and 2.6, respectively.

## 2.2 Post-validation method

The goal of this study is to evaluate whether the three LD stations during OUTPACE sampled in a homogeneous body of water. In order to achieve this goal, a number of steps were undertaken:

- Validity of application and environmental context. As mentioned in the Introduction, if a physical structure such as an eddy or front is present, its dynamics will dominate and must be taken into account. Additionally, since we used physical water properties in this study, we must determine whether biogeochemical gradients existed at smaller scales. For this purpose, we looked at remote sensing data.

- Establishment of statistical baseline. To evaluate whether station sampling remained in one water mass, the water mass itself needed to characterized. This was achieved by initializing a baseline within the timeseries of hydrographic properties. The subsequent time evolution of these properties within the defining dataset served as a first test for whether sampling stayed in one environment.

- Spatial scale determination and baseline context. If timeseries analysis showed no change in water properties, complementary data from farther away were compiled to evaluate the spatial scale at which the water mass did change. These data were also used to contextualize whether the statistical baseline over-estimated or under-estimated water mass variability.

- Currents analysis and Lagrangian risk. The spatial scale of the water mass determined, water trajectories were used to evaluate at what point the observed flow regime might have brought another water mass into contact with LD station sampling near the SedTrap drifter.

The following sections in the Methods are organized around these steps, detailing the data and analyses involved for each step.

## 2.3 Validity of method application and environmental context

Detection of physical structures and biogeochemical gradients used satellite measurements of sea surface temperature (SST), surface chl-$a$, and sea surface height with its associated geostrophic currents. These data were also used in the LD site selection phase (Sect. 2.1). All processed satellite data were provided by CLS with support from CNES. SST was derived from a combination of AQUA/MODIS, TERRA/MODIS, METOP-A/AVHRR, METOP-B/AVHRR sensors, with the daily product produced being a weighted mean spanning 5 days (inclusive) previous to the date in question. Weighting was greater for more

recent data. Similar to SST, chl-$a$ was a 5 day weighted mean produced by the Suomi/NPP/VIIRS sensor. The SST and chl-$a$ products had a $0.02°$ resolution, equivalent to $\sim 2$km. These satellite products spanned from 1 December 2014 to 15 May 2015. In order to compress the daily satellite products, weighted temporal means were calculated. For each grid location, the weight for a given day was inversely proportional to the distance from the grid point to the ship's daily position.

Temporal fluctuations of SST and surface chl-$a$ were determined by producing timeseries of both variables within a given

spatial range surrounding the starting position of the three LD stations. The spatial range consisted of a 120 x 120 km box centered at each LD station. Satellite pixels falling within this region were used to create a probability distribution function. The 120 km square size was chosen because 60 km is a typical size of the Rossby radius of deformation for the region (Chelton et al., 1998). Sudden changes in SST and chl-$a$ distributions indicated strong surface forcing or the passage of gradients, which could invalidate the applicability of the method.

Local surface currents derived from altimetry were also provided by CLS with support from CNES. These data come from the Jason-2, Saral-AltiKa, Cryosat-2, and HY-2A missions, cover a domain from $140°$ E to $220°$ E, and $30°$ S to the equator, covering the yearlong period of June 2014 to May 2015. The velocity grid had a $\frac{1}{8}°$ resolution, applying the FES2014 tidal model and CNES_CLS_2015 mean sea surface. Ekman effects due to wind were also added using ECMWF ERA INTERIM model output.

## 2.4 Establishment of statistical baseline

Water mass characterization depended upon observations of conservative temperature ($C_T$) and absolute salinity ($S_A$), or T-S measurements. The statistical baseline, which served as the reference for each LD station, needed to reflect the initial

state of the water near the SedTrap drifter. While the SedTrap drifter itself had CTD sensors onboard, these were fixed in depth and did not resolve the full variability of the water column. Additionally, although the SedTrap drifter served as the moving station's location, water derived from the shipboard CTD-rosette ultimately served as the source material for the biogeochemical measurements taken during the cruise. The shipboard casts were always positioned near the SedTrap drifter, averaging 1.2 km over the entire cruise. For these reasons the CTD cast data were chosen for the baseline calculation, while both SedTrap drifter and CTD cast data were included in the timeseries analysis.

### 2.4.1 CTD data for timeseries analysis

The shipboard CTD employed during OUTPACE was a Seabird SBE 9+ CTD-rosette, with two CTDs installed. Data from each cast were calibrated and processed post-cruise using Sea-Bird Electronics software into 1 m bins. All CTD data from other instruments mentioned later were likewise processed using Sea-Bird Electronics software. $S_A$, $C_T$, and potential density ($\sigma_\theta$) were calculated using the TEOS-10 standard (McDougall and Barker, 2011). In total, over 200 CTD casts were performed during OUTPACE. Most SD stations had three or four casts, except for SD13, which had time for only one cast owing to a medical emergency. The LDA, LDB, and LDC stations had 46, 47, and 46 casts, respectively, each approximately 3 h apart. During LDA, the two drifting moorings accidentally collided and, due to the time necessary to disentangle them, there is a gap of 9 h in the timeseries. The majority of CTD casts were to 200 m depth, with at least one 2000 m cast per station. Mixed layer depth was determined using de Boyer Montégut et al. (2004)'s method, by finding the depth where density has changed more than 0.3 kg m$^{-3}$ from a reference value, which was chosen to be the value at 10 m depth. The 10 m reference was chosen because post-processed CTD casts did not always include the surface.

The SedTrap Drifter had on board six SBE 37 Microcat CTDs. Their depths, as determined by mean in situ pressure, were $\sim$ 14, 55, 88, 105, 137, and 197 m. These instruments yielded data every 5 min during their deployments. As mentioned in the previous paragraph, during LDA, the SedTrap Drifter tangled with the Production Line, and so the data presented here from LDA came from its re-deployment until the end of LDA. No gap in the data occurred for LDB or LDC.

### 2.4.2 Tracer analysis and baseline definition

The need for a baseline within the OUTPACE dataset can be shown by comparison of the CTD data with climatologies such as the World Ocean Atlas (Fig. S1; Boyer et al., 2013). While OUTPACE observations were consistent with these previous observations, when metrics of variability were available they produced envelopes of max/min T-S values large enough to preclude distinguishing between different stations. Since no other sufficiently fine data were available to compare T-S measurements, data from within each station were used to create a reference baseline of T-S variability. Given the lack of fine variability data and the need to work within the dataset of a single cruise, another approach was needed to condense T-S variability so that physical environments can be distinguished.

Generally, over the upper 200 m of the water column, the depth range of most of our CTD casts, a given profile of T-S values will vary along a curve (Stommel, 1962). This reflects how each profile is made up of increasingly denser layers over depth, each with distinct histories. In some sense, these layers could be considered their own physical micro-environment,

and their ensemble constitutes the physical environment. Assuming that the density layers were not subject to strong forcing such as diapycnal mixing events or atmospheric effects, their values should have remained constant until isopycnal exchange or diffusion could occur over longer timescales. Treating these density layers as separate entities, variations of T-S along isopycnal surfaces can provide an approach to distinguish physical environments, the goal of our analysis.

Using density as one variable, another is needed to fully describe a water parcel's characteristics, ideally one which is independent of density. Spice, a variable constructed from T-S, is well suited for this purpose. Spice is defined such that hot and salty water is 'spicy', a convention dating to Munk (1981). In the formulation proposed by Flament (2002), its isopleths are everywhere perpendicular to isopycnals, and it effectively both encapsulates and accentuates T-S variability at a given density into a single value. Therefore, in our analysis, spice variability in a given density layer was used to define the statistical

baseline, and determine whether a physical environment changed during OUTPACE.

     The statistical baseline was defined as a functional fit between $\sigma_\theta$ and spice measurements at the beginning of each LD station in the upper 200 m of the water column spanning the euphotic zone. The period of time used for defining the baseline was chosen to be the local inertial period, so that internal wave effects would be minimized. For each station, this meant that the first 13, 15, and 15 casts were used for LDA, LDB, and LDC, respectively. A regular grid of density values was created,

with one-fourth the number of values as the total number of observations. The fit of baseline spice, or $S_{base}(\rho)$, was calculated inside a moving window of $\pm\,0.1$ kg m$^{-3}$, along with the standard error in spice, $SErr_{base}(\rho)$. Only values corresponding to windows with at least 50 observations were kept.

     Comparisons between the baseline and new $\sigma_\theta$-spice measurements were made using a Z-Score, following the general formula

$$Z(\rho_{obs}) = \frac{S_{obs} - S_{base}}{SErr_{base}} \tag{1}$$

where, for a density observation $\rho_{obs}$, $S_{obs}$ is the observed spice, with $S_{base}$ and $SErr_{base}$ being the linearly interpolated functional baseline spice value and standard error. The assumption applied in this analysis is that while a continuous curve in T-S, or $\sigma_\theta$-$S$, is to be expected and can be fit to a function, the isopycnal layers were independent of each other, and represented different physical sub-populations. Keeping track of variability through Z-score tied to a functional $\sigma_\theta$-$S$ relationship produced

a flexible metric. For sensors fixed at a certain depth, such as for the SedTrap drifter, a Z-score could be computed irrespective of whether internal waves were shoaling or deepening isopycnals.

## 2.5   Spatial scale determination and baseline context

The Z-scores derived from the CTD and SedTrap timeseries provided a first-order evaluation of physical variability in the immediate surroundings of the SedTrap drifter as it moved through time. If large variability (|Z|>2, in the traditional $\alpha = 0.05$)

was observed, then the physical environment likely had changed. If |Z|<2, however, this was not a guarantee that the physical environment had not changed. Since the functional fit of $\sigma_\theta$ and spice was based only upon the data from OUTPACE, Z-score is a relative measure of variability. In order to test whether the $\sigma_\theta$-spice relationship was robust, it was necessary to extend the Z-score analysis farther in space to include complementary density and spice measurements. If Z-scores remained low for

large distances, then the $SErr_{base}$ was too large. By compiling independent Z-scores over larger distances, we can test whether there is a relationship between Z-score and distance.

During OUTPACE, complementary $\sigma_\theta$-spice observations stemmed from neighboring stations, the SD stations (Fig. 1). Additionally, surface measurements for the entire cruise were provided from a Seabird SBE 21 SeaCAT Thermosalinograph (TSG), with SBE 38 thermometer using the ship's continuous surface water intake. Subsequent to post-cruise processing of TSG data as detailed in Alory et al. (2015), the timeseries was available in 2 min intervals.

The relationship between Z-score vs. distance was used to evaluate the baseline. Distances were calculated from the ship position of observation and the initial CTD cast position for the LD station. For the SD stations, the Z-scores found by the functional fit of spice for each cast were binned by density, in a regular grid with bins of 0.25 kg m$^{-3}$ width. TSG data from during the LD stations were excluded, and Z-scores were binned by distance from the station, in 10 km increments for the first 100 km, and then 20 km afterwards until 500 km. The spatial scale $R_Z$ was determined where Z$\geq$2, and used to evaluate the ability of the statistical baseline to discern gradients in physical properties. A natural spatial scale to serve as a useful reference to the empirical distance is the first Rossby radius of deformation, approximated via Wentzel-Kramers-Brillouin (WKB) method by Chelton et al. (1998) as

$$R_D = \frac{1}{\pi f} \int_{-H}^{0} N(z)dz \qquad (2)$$

where $f$ is the local coriolis parameter and $N(z)$ is the depth-dependent Brunt-Väisälä frequency. $R_D$ was calculated for each LD station using the deepest cast available: 2000 m casts for LDA and B, and a deep 5000 m cast for LDC. $N$ was calculated with centered differences of the 1-m binned density profiles.

## 2.6 Currents analysis and lagrangian risk

The previous step analyzed the relationship between Z-score and distance, providing an estimated distance over which the physical environment changed. In order to evaluate the risk that the SedTrap drifter encountered different water masses, an analysis of the local currents was needed. Since it is clear that the SedTrap drifter was not perfectly Lagrangian, and that vertical shear could transport layers at different rates, it was necessary to see if water at specific depths could have advected the distance over which different water masses appear.

The in situ velocities for each LD station were derived from the shipboard acoustic Doppler current profilers (SADCP), two Ocean Surveyors at 150 kHz and 38 kHz. Timeseries data for the SADCPs were post-processed using the CASCADE software package (Le Bot et al., 2011; Lherminier et al., 2007) and binned into 2 min intervals. The 150 kHz SADCP provided a depth resolution of 8 m, with bins starting from 20 m, and reliable data coverage down to 200 m depth. Since the SedTrap Drifter had sediment traps extending down to 500 m depth, the 38 kHz data was also used, albeit with reduced depth resolution of 24 m bins, extending from 52 m down to 1000 m. Additional in situ velocities came from six Nortek AQUADOPP current meters, positioned at 11, 55, 88, 105, 135, and 198 m on the SedTrap Drifter. The post-processed AQUADOPP timeseries provided observations every 5 min.

Velocities were integrated using a first-order Euler method to calculate the theoretical trajectories of water subsequent to the beginning of each LD station. Since the object of these calculations was to see whether water could have traveled a critical spatial scale, for each dataset the maximal amount of time was given for the time integration. SADCP timeseries spanned between the first and last CTD of the LD station, using the ship position as the initial position. The AQUADOPP integrations spanned the entirety of valid data and used the corresponding SedTrap Drifter satellite fix for an initial position.

To compare the integrated velocity positions with the realized positions of the SedTrap drifter and SVP drifters, GPS positioning was achieved by use of Iridium telemetry. Positions were successfully found for LDA before and after the SedTrap Drifter's re-deployment, along with all of LDB. During LDC, the battery of the positioning antenna ran out and so the timeseries for LDC positions of the SedTrap Drifter was shortened. Since only the initial position is needed for the velocity integration, the AQUADOPP integration was continued beyond this positioning failure until the SedTrap Drifter was recovered. Positions of the SVP drifters deployed at each station were successfully retrieved for all three LD stations. Satellite fixes were available spaced about 1 h apart for both datasets. Both SedTrap Drifter and SVP positions were interpolated to hourly timeseries. SVP positions were used to compute relative dispersion (supplementary Fig. S2) using the definition for N particles (LaCasce, 2008),

$$D(t) = \frac{1}{2N(N-1)} \sum_{i \neq j}^{N} [x_i(t) - x_j(t)]^2 \tag{3}$$

where $N$ here is the total number of SVP drifters, and $x$ the timeseries of the drifter $i$'s $x, y$ position.

## 3   Results

### 3.1   Satellite data, temporal context, and method applicability

The regional distributions of SST and surface chl-*a* as seen during the OUTPACE cruise are shown in Fig. 1. The data in Fig. 1 are weighted means, with the weight being the inverse square of the ship's daily distance to each pixel. A north-south meridional gradient was found in SST, with warmer water near the equator ($\sim 30°$C) and cooler water poleward ($\sim 25°$C). This gradient was uninterrupted for the duration of the OUTPACE cruise. Due to the zonal shiptrack the surface thermal conditions observed by the ship during OUTPACE were relatively homogeneous. Furthermore, no strong temperature gradients, indicative of frontal or eddy structures, were visible in the vicinity of the stations. While a north-south regional gradient was found in SST, the opposite was found in chl-*a*. Chl-*a* values were around 0.3 mg m$^{-3}$ in the western portion of the domain, west of 190° E. Stations LDA and LDB were in this region, with LDB positioned inside a bloom with values near 1 mg m$^{-3}$. More details concerning the LDB bloom can be found in de Verneil et al. (2017). Chl-*a* values dropped precipitously, over an order of magnitude to 0.03 mg m$^{-3}$, just east of LDB near LDC. The low value of chl-*a* was indicative of the South Pacific Gyre (SPG; Claustre et al., 2008).

Since SST was relatively unchanging during OUTPACE, Fig. 2 provides zoomed-in views of the chl-*a* data for the three LD stations, with domains chosen to include the nearest SD stations. The spacing of the SD stations was relatively regular along

the OUTPACE transect (Fig. 1b). In Fig. 2a the enhanced chl-*a* was distributed evenly inside the domain, so no clear surface gradients are present. In Fig. 2b, the chl-*a* was concentrated in the aforementioned bloom, with values higher than those seen in Fig. 2a near LDA. The size of the bloom was large enough to cover most of the 120 km x 120 km region shown in Fig. 2b, so the bloom edges were far away from station LDB's initial position. By contrast, waters outside the bloom had chl-*a*

values lower than in Fig. 2a. The low chl-*a* values near LDC in Fig. 2c were typical of the SPG, and no visible patches of chl-*a* indicated sharp gradients.

The timeseries of chl-*a* and SST for the three stations are shown in Fig. 3. Comparing the three LD stations, a few patterns emerge. SST showed similar trends across the three LD stations. All stations experienced warming trends from December 2014 to mid-March 2015, consistent with summer heating. The lack of data from cloud cover sometimes led to abrupt drops in

the distribution of daily SST shown. The timing of maximum temperature, and the magnitude of that warming, however, did differ between LD stations. A rapid heating in December 2014 occurred around LDA's position, which then slowly continued until early March 2015, at which point temperatures began to drop. Towards the end of sampling at station LDA the SST rises, indicating possibly another warming event occurred or the arrival of a warm patch of water. Depth-resolved application of our method in the later sections will evaluate this possibility. The overall evolution in LDA's temperature during the period shown,

from ∼26 to 30° C, represented a 4° change. LDB showed a slight cooling in December 2014, but this may have been an artifact of cloud cover. Station sampling for LDB occurred immediately after the maximum heating, though the values seen at LDB were relatively stable and slightly warmer than at LDA. The maxima in temperature for LDA and LDB seemed to overlap in time, in early March 2015. LDB's change in temperature, from ∼27 to 30° C, was a 3° change. LDC had the smallest change in temperature, from ∼27 to 29 °C for a 2° change. Sampling for LDC coincided with the warmest period observed in

the satellite data, in late March 2015, and was stable for the LD sampling period.

The timing of temperature maxima is important to note for biological reasons, since $N_2$ fixation by *Trichodesmium* spp. is known to occur in warm, stratified waters (specifically, a ∼25° C threshold, White et al., 2007), and one of the goals of OUTPACE was to observe this biogeochemical process. Since SST was above 25° C for all stations throughout this period, the thermal conditions during OUTPACE would not have limited $N_2$ fixation.

In between December 2014-January 2015, the region around LDA had higher chl-*a* concentrations than LDB. The period between February and May 2015 showed a remarkable increase in chl-*a* near the LDB site. This was due to advection of the surface bloom, which subsequently collapsed and advected away, as documented in another study in this special issue (de Verneil et al., 2017). The downward trend of chl-*a* during this period is more indicative of in situ evolution, rather than advection of the bloom's boundaries, and does not invalidate subsequent use of our method. Near LDC, chl-*a* was systematically

low, a reflection of the goals of OUTPACE to sample in the SPG.

Besides the increase in SST at the end of LDA and the decrease in chl-*a* during LDB, both SST and chl-*a* for the LD stations were stable, providing evidence that no surface gradients, physical or biological, immediately invalidate the application of our strategy. Though the change in chl-*a* at LDB has been argued to be due to endogenous dynamics in the aforementioned study, application of our post-validation method provides an independent test of whether advection of gradients could be responsible.

Likewise, the method will also determine whether the surface increase in SST during LDA was reflective of changes at depth.

As a final note, Rousselet et al. (this issue) found with satellite-derived surface velocities that during LDC a coherent structure was present, despite the lack of surface SST and chl-*a* gradients. Since tracer gradients are non-existent at the surface, we find that this is not a strong structure, and does not invalidate the applicability of our approach; rather, the application of depth-resolved in situ measurements of tracers and velocities will serve as an independent evaluation of this finding.

## 3.2 In situ properties, statistical baseline, and timeseries analysis

The hydrographic variability during the three LD stations and surrounding SD stations are shown in the T-S diagrams of Fig. 4. All three stations followed a general pattern, where surface water near the 1022 kg m$^{-3}$ isopycnal and 29° C temperature (though LDB had warmer surface water, Fig. 4b) dropped in temperature and increased in salinity until a subsurface salinity maximum near the 1025 kg m$^{-3}$ isopycnal. The increase in salinity maximum from LDA to LDC reflects the high salinity tongue of the South Pacific (Kessler, 1999). The surface water in LDA (Fig. 4a) showed a bifurcation. This change was manifest in the satellite data timeseries, as well. Whether this is due to a heating event or the arrival of new water at the end of LDA will be addressed in the timeseries analysis below. For LDA, neighboring stations SD2, 3, and 4 largely overlapped with the LDA profile. SD3, the station closest to LDA, almost entirely overlapped the LD profile, except for a subsurface salinity deviation below the 1024.5 kg m$^{-3}$ isopycnal. SD2 and SD4 showed greater deviations, with SD4 being saltier than LDA for almost the entire profile. Similar overlaps occurred with LDB and its surrounding stations, SD12 and SD13 (Fig. 4b). SD12 showed lower salinity near the surface, with a kink in salinity at the 1025 kg m$^{-3}$ isopycnal. The salinity offsets of SD4 and SD12 at depth are within climatological variability (Figs. S1 and S2). SD13 had similar surface structure to LDB, but higher salinities from the 1023.5 to 1025 kg m$^{-3}$ isopycnal. The LDC, SD13, and SD14 (Fig. 4c) profiles nearly entirely overlapped except near the surface when the SD stations were at first less salty at the surface and then became more salty. Additionally, the saltier nature of LDC relative to LDA and LDB, especially between 1024 and 1025 kg m$^{-3}$, was visible. The variability in T-S values between stations was within the range seen in the climatology of the region (Figs. S1 and S2).

The LD statistical baselines of spice in density space, with means and intervals of two standard errors, are shown in Fig. 5. These standard error intervals, representing the inherent variability in the baseline, show the values wherein a Z-score of $\leq 2$ was achieved. LDB and LDC overlapped for essentially their entire profiles. All stations are missing observations near the surface and mixed layer due to the intense stratification which left several density bins with less than 50 observations, the threshold used in the spice analysis. LDA was noticeably less spicy than the other two LD stations for density less than 1024 kg m$^{-3}$. At the highest densities, all three LD stations overlapped. The envelope of two standard errors, or Z-score $\geq 2$, show that variability has some dependence on depth. The LDA baseline shows high variability near the surface, with a thin envelope below down to 1024 kg m$^{-3}$, and widening at depth down to 1025 kg m$^{-3}$ and beyond. LDB did not have high surface variability, but the envelope widens shortly below 1023 kg m$^{-3}$. Variability in LDC shows similar widening as in LDB, with a noticeable pinch in the envelope near 1024.5 kg m$^{-3}$.

The Z-score timeseries for the SedTrap Drifter sensors are shown in Figs. 6-8. During LDA, at 14 m depth (Fig. 6a), after the inertial period baseline determination the Z-score first descended, increased, and then leveled off after the first two and a half days. Afterwards, the Z-score increased, reaching 2, decreased, and surpassed Z=2 before falling again. The SedTrap drifter at

m showed no trend (Fig. 6b), and a single Z-score was seen below -2. Z-scores for 105, 137, and 197 m (Fig. 6d-f) showed no temporal trends and were always less than 2 in magnitude. The timeseries at 88 m showed no trend, but the variability in Z-score increased time, with some observations surpassing |Z|=2.

LDB SedTrap drifter Z-scores (Fig. 7) showed similiar patterns to LDA. The surface sensor (Fig. 7a) decreased and increased over the first two days, then leveled with temporary departures below -2. The sensors at 55, 105, 137, and 197 m (Fig. 7b,d-f), similar to LDA, showed no trends and low variability. A few observations below -2 occurred at 55 m. The Z-scores at 88 m showed no trend, similar to LDA, with enhanced variability and some |Z|>2, but no time-dependence.

The LDC Z-scores were large at more depths than the other LD stations (Fig. 8). Values at 14 m started with Z>2, but that dropped before rising again after a day, before slowly dropping and eventually decreasing to $\sim$-2 at the end of LDC. Z-scores at 55, 137, and 197 m showed no trends in Z-score, and had limited observations with |Z|>2. At 88 m, no trend was seen, and for the first two days there were few observations with Z<-2. Toward the end of LDC, two spikes with Z>2, with Z$\sim$4-5, occurred with returns back to |Z|<2. The Z-scores at this depth ended near Z=2. Observations at 105 m started around -2<Z<0, but spikes with Z$\sim$2 occurred. Over time, Z-scores trended upward with more oscillations, with a shift to Z>2 becoming dominant during and following March 27, 2015.

CTD Z-score timeseries are shown for LDA, LDB, and LDC in Figs. 9, 10, and 11, respectively. LDA Z-scores were generally |Z|<2, but Z-scores for densities $\sigma_\theta < 1022$ kg m$^{-3}$ were greater than 2 starting March 1, and continued for the rest of the station. The increasing trend in Z-score near the surface was also reflected in the SedTrap drifter. LDB CTD Z-scores showed almost no observations with |Z|>2. These occurred at the surface with low densities, and a few near $\sigma_\theta \sim$1023.25 kgm$^{-3}$. All these observations occurred before or around March 19, and no temporal trend in |Z|>2 was seen. The Z-scores for LDC showed similar trends to the SedTrap drifter. Near the surface close to $\sigma_\theta \sim 1022$ kg m$^{-3}$, |Z|>2 was seen early in the timeseries, but then dropped to |Z|<2 until another increase around March 27. This pattern was similar to the SedTrap drifter's observations at 14m. Regions of |Z|>2 appeared in the 1024-1025 kg m$^{-3}$ range, primarily during March 27. A small density range near 1024.5 kg m$^{-3}$ showed |Z|>2 during March 26, but as time went on a larger swath of density had |Z|>2, and this change was largely permanent. Near 1025 kg m$^{-3}$, a separate series of large Z-scores appeared on March 27 and lasted for most of the rest of LDC.

### 3.3 Spatial scale and baseline context

The TSG Z-scores for the three LD stations are shown in Fig. 12. For LDA, Z>2 occurred at 150 km. Z-scores were consistently large farther away from this point. The LDB TSG surpassed Z=2 at 55 km, though Z-score diminished again 300 km away. For LDC, TSG Z-score reached 2 at 35 km distance, and |Z| oscillated between larger and less than 2 farther away. Therefore, at the surface layer, 150, 55, and 35 km were the spatial scales. Since at least some Z-scores were found to be greater than 2, the baseline was sensitive enough to determine gradients over a 500 km scale. Since Z-scores for LDB and LDC were not consistently |Z|>2, then the baseline's sensitivity was perhaps not as great as LDA. The Rossby radii for the three stations were 46.5, 48.8, and 60 km. The spatial scales for the TSG data at LDB and LDC matched up to the Rossby radii, whereas LDA's TSG data indicated a larger scale.

Z-scores from the SD stations are presented in Fig. 13. For LDA, the |Z|>2 distances demonstrated density dependence. Near 1022 kg m$^{-3}$, |Z|>2 immediately, at ~45km, though this did not occur at the surface. Approaching 1000 km distance, |Z|>2 occurred from the surface to 1024 kg m$^{-3}$. By 3500 km, all density layers show |Z|>2. LDB showed large Z-scores in some density layers at the closest SD station located 189 km away. Past 750 km, Z from 1022-1024 kg m$^{-3}$ was consistently high.

For densities greater than 1024 kg m$^{-3}$, Z-scores were enhanced between 1000 and 1500 km, but then decreased farther away. LDC's Z-scores show that |Z| was greater than 2 from the first observations at 310 km. All density layers showed enhanced Z values, with the majority of all observations being larger than 2. Similar to the TSG data, the SD stations showed that the baselines were sufficiently sensitive to detect physical gradients on large scales, with some detecting changes immediately. Putting together the near-surface TSG data and CTD data from the SD stations, LDA showed smaller |Z|=2 scales at depth, whereas LDB and LDC both showed variability both near the surface and at depth at smaller scales. In order to be the most conservative in our velocity and trajectory estimates, we will use the smallest spatial scale of |Z|=2 to determine the spatial scale $R_Z$ and evaluate Lagrangian risk, namely 45 km for LDA, 55 km for LDB, and 35 km for LDC.

Having evaluated the ability of the statistical baselines to sense physical gradients over large scales, we are now ready to analyze the currents and trajectories.

## 3.4 Velocities and lagrangian trajectories

Timeseries of the 38 kHz SADCP and AQUADOPP data are presented in Fig. 14. The LDA timeseries of SADCP u and v components (Fig. 14a,d) showed strong near-inertial oscillations in the upper 200 m, with velocities reaching magnitudes of 0.6 m s$^{-1}$. A weaker tidal component was also present in this layer: below 200 m, vertical columns of alternating velocity sign indicated the semi-diurnal tide. These tidal signatures were also the dominant signal in the LDB and LDC timeseries (Fig. 14b-c,e-f). The mixed layer, which, for most of the cruise, was ≤20 m, was not resolved by either SADCP. So, the near-surface velocities were only captured by the 11 m AQUADOPP and the SVP drifters drogued at 15 m. Comparing the 55 m AQUADOPP timeseries with the 52 m SADCP, the two data sources displayed similar trends for LDA. The strong near-inertial oscillations led to correlations between the AQUADOPP and 38 kHz timeseries of 0.75 and 0.76 for the u and v components, respectively. During LDB and LDC, the weaker currents did not correlate as well, leading to u,v correlations of -0.0137, -0.0554 (LDB), and 0.30, 0.37 (LDC), respectively. For comparison, the 150 kHz 52 m timeseries produced u,v correlations with the AQUADOPP of 0.83, 0.80 (LDA), 0.00, 0.02 (LDB), and 0.68,0.68 (LDC). Vector correlations using the method of Crosby et al. (1993) for the three timeseries (not reported) similarly showed a maximum for LDA, minimum near-zero for LDB, and low values for LDC. These differences likely result from higher frequency fluctuations of the currents, at the inertial and tidal frequencies. The fact that a higher correlation is obtained at LDA is probably partly the consequence of the larger horizontal scales of the near-inertial signal dominant at LDA compared to the baroclinic tidal signal, e.g. resulting from the dispersion relation (Alford et al., 2016). These oscillations, and their implications for turbulent mixing, are analyzed in greater detail in Bouruet-Aubertot et al. (this issue).

The disagreement between the two velocity data sources had an impact on the integrated trajectories. Take, for example, a closer inspection of the SADCP and AQUADOPP during LDA, which had the strongest currents. The initial positions of the

ship and the SedTrap Drifter were 1.46 km apart. After 3 days and 2 hours, the AQUADOPP integration had traveled 67.75 km, the SADCP 60.71 km, with a final separation of 10.89 km. The result was a positional drift of $\sim3$ km day$^{-1}$, or an average increase in position difference of 147 m for each km traveled. A similar analysis for the LDB timeseries, with weaker currents but essentially no correlations over 4 days and 15 hours, resulted in a positional drift of 3.19 km day$^{-1}$, with an increased

position difference of 318 m for each km traveled. Thus, a lower correlation timeseries, but with lower magnitudes, resulted in similar misfit in the integrated trajectory.

The trajectories of the integrated velocities, as well as observations of SedTrap Drifter and SVP positions, are presented in Fig. 15. The average altimetry-derived currents suggested there should be recirculation around the positions of LDA and LDC, whereas LDB had a mean northward flow (Fig. 15 a-c). The SedTrap Drifter trajectory for LDA did not follow the surface

altimetry currents and their anticyclonic flow, but instead underwent several oscillations while cruising in a west-northwest direction (Fig. 15a). The SVP drifters for LDA (Fig. 15g), while also undergoing oscillatory loops, instead drifted to the south. The 38 kHz SADCP velocities showed a transition over depth with shallow water moving south-southwest, but with increasing depth the trajectories flowed northwest in a similar fashion to the SedTrap Drifter. During LDB, the SedTrap Drifter went north-northeast, in agreement with the altimetry currents (Fig. 15b). The SVP drifters moved in a similar fashion, north-northeast,

though they ended up undergoing more oscillations and eventually advected more eastward (Fig. 15h). The 38 kHZ SADCP velocities demonstrated that shallow depths flowed east like the SVP drifters, but with depth this advection swung to a more northerly direction (Fig. 15e). The LDC SedTrap positioning was relatively uninformative, since the Iridium satellite fix was unavailable for the second half of the LD station and so showed little displacement (Fig. 15c). The SVP drifters for LDC (Fig. 15i), similar to LDB, underwent several oscillations and were advected the farthest, moving in a southeast direction. The

SADCP data showed a shallow flow to the east, similar to the SVPs, but with depth the majority of trajectories oscillated near the station and even flowed southwest.

For all the LD stations, the SedTrap Drifter stayed within a radius of $R_Z$ and $R_D$ centered at the LD starting position. Integrated velocities of the SADCP also stay within $R_Z$ and $R_D$, except for the trajectory nearest the surface for LDA. The SVP drifters for LDA and LDB also stayed within the $R_Z$ and $R_D$ distances, though the LDB SVP trajectories come close

to $R_Z$. For LDC, however, the SVP drifters traveled farther than $R_Z$, but shorter than $R_D$ away from the initial position. $R_D$ was larger than $R_Z$ for all three stations. Since surface SADCP data for LDA crossed $R_Z$, we evaluate that the water during LDA might be from a different water mass. Likewise, the SVP trajectories for LDC crossed $R_Z$, meaning that surface water for LDC might be from a different water mass at the end of the station. As seen in the timeseries analysis, water at depth for LDC also changed part-way through the station, though the SADCP velocities show little displacement. As a result, according to our

protocol the water observed in these density layers may have derived from a new physical environment, and biogeochemical observations at the end of LDA (primarily the surface) and LDC (surface and at depth) may need to be examined in closer detail for changes.

We conclude by noting that the trajectories highlight how water seen during the LD stations derived from farther away than the SedTrap drifter's position, as well as from differing directions. In effect, the drifter is only quasi-Lagrangian. Beyond the

layers shown to have different water masses due to the timeseries analysis, the conservative length scale $R_Z$ helps determine

the maximum spatial footprint where sheared layers can be said to be from the same water mass. The results suggest, therefore that LDB's layers were all from a single physical environment, surface observations for LDA and LDC were suspect towards the end, and LDC water in the 1024-1025 $\mathrm{kg\ m^{-3}}$ range should be closely examined after March 27, 2015.

## 4  Discussion

### 4.1  Tracer analysis and spatial scale determination

The main goal of this study is to determine whether the quasi-Lagrangian sampling strategy during the LD stations of OUT-PACE was successful. Since the SedTrap drifter cannot be truly Lagrangian, successful sampling is judged by whether observations stem from a single physical environment. The motivation behind this exercise is to independently evaluate if the biogeochemical measurements of OUTPACE represented a single biogeochemical milieu, rather than the advection of the Sed-Trap drifter into a different area, as well as the advection of different water toward the drifter. Evaluation of the strategy is grounded in the variability of T-S and analysis of water velocities.

Before the spice analysis was conducted, the initial context of SST and chl-*a* variability at the surface in space and time was provided by satellite products. At the regional scale of the WTSP, the LD stations were roughly positioned within the zonal gradient of chl-*a* and meridional gradient in SST (Fig. 1). The gradients of surface chl-*a* around the LD stations were minimized in relation to the regional-scale gradients, partly by design in the process of choosing station locations (Fig. 2). The temporal trends of SST and chl-*a* largely reflected the seasonal cycle: chl-*a* was decreasing at the end of the summer in the MA and low values dominated in the SPG; SST reached its peak due to late summer timing (Fig. 3). The timing of temperature maxima is important to note, as mentioned in Sect. 3.1, since $N_2$ fixation by *Trichodesmium* spp. is known to occur in warm waters, and one of the goals of OUTPACE was to observe this biogeochemical process. While these satellite data are sufficient to identify large-scale structures or temporal trends, the LD stations by design were in regions where it is difficult to judge whether the SedTrap Drifter stayed in the same water mass from these surface data alone. However, they do justify the use of our methodology, ie there were no strong mesoscale structures in the vicinity, and no nearby chl-*a* gradients were present. As mentioned previously, the use of remote sensing data to help identify small-scale surface structures during OUTPACE is further explored in Rousselet et al. (this issue). In that study, a possible coherent structure was found during LDC, though this derives from satellite-derived surface velocity data; the SedTrap drifter's Lagrangian limitations preclude testing whether the structure was truly coherent. To continue with the post-validation strategy evaluating LD sampling, in situ data are now needed.

The depth-resolved in situ T-S data (Fig. 4) helped to capture some of the variability present during the LD and surrounding SD stations. The T-S structures showed consistent values during LD stations with deviations observed in the neighboring SD stations. As with the satellite data, the in situ data in this form, although informative, provide qualitative interpretation. In order to identify water masses, traditional quantitative methods require identified water masses (Mackas et al., 1987; Poole and Tomczak, 1999). In some regions of the ocean, these methods are difficult to apply. Also, maybe the full complement of tracer data (dissolved $O_2$, nutrients, etc.) cannot be used (like in this study) because they are liable to rapidly change in the euphotic zone due to biogeochemical processes. Additionally, local mesoscale activity can contribute to variability, as has been seen

in the WTSP (Rousselet et al., 2016). In that study, $O_2$ measurements were the distinguishing tracer, which we are precluded from using. As a result of all these factors, another quantitative method that works within the dataset of a single cruise in the WTSP is needed.

The quantitative approach used in this study leverages the large quantity of in situ T-S data available from multiple platforms to condense the physical variability present in the WTSP during OUTPACE over 4000 km distance during austral summer 2015. In order to do this, the statistical baseline in spice was defined (Fig. 5). In effect, as opposed to an absolute measure (i.e. specific water mass determination), this provides a relative measure of variability that can be used. The timeseries analysis for the latter portion of the SedTrap drifter and CTD timeseries data showed density layers where variability was enhanced, and caution should be applied to the analysis of biogeochemical data. Since the baseline is a relative measure, observations from outside the LD station were used to see at what scales physical gradients appear. The relationship between distance and variability (summarized as Z-scores) provided a method by which to establish this scale. Overall, variability increased with distance, as one would logically expect, but this was not monotonic across all datasets. Therefore, the first increase in Z-score above 2 (using an $\alpha = 0.5$ criterion) was used, and the smallest of these scales across datasets and density layers was conservatively chosen in determining the cut-off scale $R_Z$. These distances were of the same order of magnitude as the Rossby radius $R_D$.

Despite the testing of the baseline on complementary data outside of LD sampling, there is still the question of its generality. In some instances, the variability can be partly attributed to an in situ process, such as the near-surface changes in LDA coinciding with surface heating (though advection possibly played a role, as seen in the SADCP trajectories). While seeing increased variability with distance was reassuring, the non-monotonic nature of some Z vs. distance relationship raises some questions. Is the water on the other side, where |Z| goes back to below 2, truly the same water mass? Is |Z|>2 truly a change in water mass relative to traditional methods? Is the Z-score approach based on the spice-density relationship more, or less, sensitive than these methods? These questions merit further study, and will have to be explored using both methods simultaneously.

The example of surface heating at LDA introducing unwanted variation brings up another assumption in our analysis: we used spice hypothesizing that there was no diapycnal forcing. Clearly, at the surface, atmospheric forcing can influence the water's T-S (and spice) characteristics, and so will impact TSG measurements as well as observations in the upper mixed layer. Future applications of this method will have to take this variability into account, and perhaps make greater use of survey data to fill in the spice variability below the surface at these spatial scales. At depth, however, the greatest source of along-isopycnal gradients in T-S, i.e. density-compensated features, is mesoscale stirring (Smith and Ferrari, 2009), and so, generally, the assumption should be applicable. Part of our assumptions in developing this method requires that sampling is not near mesoscale fronts and eddies. However, their residual effects are the most likely to affect along-isopycnal variability in a given field campaign, and it is probably the reason that both Z-scores (especially near the surface) begin to increase and the $R_Z$'s were found to be at or near the Rossby radius $R_D$ for each LD station. Granted, if ship sampling happens to be placed immediately next to a strong eddy or filament, Z-scores could increase over much shorter distances, though as previously stated this situation was expressly avoided. Therefore, in situations where our starting assumptions are met, at first order the Rossby radius $R_D$ serves as a default scale at which the integrated history of previous mesoscale stirring will on average manifest itself.

In situ data and further analysis is needed, however, to verify whether smaller-scale variability is large through determination of $R_Z$, as evidenced by the appearance of salty, spicy water in the mid-water column at the end of LDC. One recommendation that emerges from these results is that, if future field measurements are to take place in an area devoid of obvious mesoscale structures, the Rossby radius $R_D$ quickly calculated from a deep CTD cast may be useful in determining at which point a 5 quasi-Lagrangian drift array should be recovered as a precautionary measure.

The choice of spice as a variable, though useful, is not a magical transformation in itself. The similarity in T-S between LD stations is still manifest in spice-density space (Fig. 5), especially at depth with some density layers overlapping. This means that for these density ranges, the stations are not distinguishable from one another. The overlap in statistical baselines further emphasizes the need to compare with other datasets to highlight and determine the scales at which variability exists. Having 10 enough data to span a sufficient spatial range is what will determine what differences in spice are relevant or not. The fact that the SedTrap drifter, positioned close to the CTD timeseries, mirrored the same trends, corroborated both the presence or lack of small-scale gradients shown through Z-score. If the SedTrap Drifter timeseries, representing the smallest scales, had displayed much larger Z-scores, then the conclusion would have been that there was large variability right next to the ship somehow missed by the CTD baseline. Instead, by showing similar trends, the SedTrap data validated the baseline and the spatial scales 15 at which the LD CTD timeseries was sampled, even if statistical baselines between the far-flung LD stations overlap.

Having considered some of the caveats and assumptions implicit in our present approach, we feel that its application for the OUTPACE campaign was warranted and subsequently validated by the consistency of the results, both in consideration of the multiple data sources concerned but also the theory of mesoscale circulation. In our application, a conservative $R_Z$ was used based on surface data at intermediate scales. Ideally, future applications of this method could hopefully use depth-resolved 20 measurements to see when surface layers diverge as opposed to deeper ones, which might be reflective of different turbulence regimes such as the submesoscale.

## 4.2   Integrated velocities and drifter trajectories

The analysis of the integrated in situ velocities, alongside the SedTrap Drifter and SVP positions, was used to determine whether certain parcels of water advected far enough during each LD station's sampling to move beyond $R_Z$, the conservative 25 spatial scale. The conclusion for the OUTPACE cruise is that almost all of these proxies for Lagrangian pathways did not, and hence the changes observed in the upper water column during the LD stations were probably due to biogeochemical processes alone. Nevertheless, given the multiple sources of trajectories available, there is no clear interpretation as to which of them is the most 'truly Lagrangian.'

As stated in the introduction, the quasi-Lagrangian drift array, spanning the top 100's of meters of the water column, with 30 various instruments having varying degrees of drag strewn at different depths, will never be purely Lagrangian. Unless a drift array is deployed in a flow that is entirely barotropic, there will likely be sources of shear in the velocity field from the mesoscale flow and from baroclinic near-inertial waves and internal tides (see Bouruet-Aubertot et al., this issue). By definition, vertical shear leads to different velocities at different depths. The net effect of these velocities results in the drift array's observed trajectory, with water at some depths moving slower or faster than the array itself. As a result of this inevitable

decoupling of in situ and drifter velocities, the water passing the drifter can come from entirely different areas. Take, for example, the near-inertial oscillations during LDA (Fig. 7a,d). The oscillations influenced the top 200m, so immediately the top sediment trap at 150m experienced vigorous currents whereas the bottom two at 250 and 500m did not. This complicates the interpretation of where the falling organic matter comes from. Strong vertical structuring of the phytoplankton community

further complicates this picture. In oligotrophic areas, light-adapted organisms are found near the surface with dark-adapted organisms near the deep chl-$a$ maximum (not to mention the near-surface $N_2$ fixation occurring during OUTPACE). During LDA, the shear resulted in bands of instantaneously opposing velocities in layers about 50 m thick. Therefore, organisms sampled by a CTD near the chl-$a$ max at 80 m would be traveling in a direction opposite to the organisms at the surface. Immediately, it seems, the expectation of being Lagrangian is lost. However, whether this is an irrevocable loss depends upon

the nature of the motions: are they associated with internal waves, such as LDA, or not? If shearing is wave-induced, then after one period, water should return to its original position. Thus, sampling over at least one wave period and within corresponding horizontal wavelengths could help to preserve the physical environment. If motions are not related to oscillatory processes, then the only recourse would be that biological communities (and the state of biogeochemical processes underway) are contiguous enough in horizontal extent that they reflect the physical spatial scales found in the tracer analysis of this study.

The potential ability of individual shearing layers to remain coherent over long timescales during OUTPACE is reflected by the SVP drifters during LDC. While they had traveled far, surpassing $R_Z$ and near $R_D$ by the end of sampling, they remained close to each other. This is also reflected in the SVP relative dispersion timeseries (Fig. S2). Typically, relative dispersion increases in the first few days in an exponential fashion, until the drifters begin to be influenced by mesoscale structures (LaCasce, 2008). For the OUTPACE SVP drifters, the exponential phase (a linear increase in log-space) did not abate in this

timeframe. However, the exponential growth rate was so low that it took over a week for the relative dispersion to arrive near $1 \times 10^8$ m$^2$. Thus, while the SVP drifters during LDC advected past $R_Z$ and near $R_D$, their spread was less than this, indicating that the surface layer in LDC was more coherent than comparing between integrated trajectories over different depths would suggest.

In addition to the sheared layers advecting past the drift array, what determines the trajectory of the array itself is difficult

to establish. During LDA, with vigorous currents in the upper 200m, the SedTrap Drifter advected north-west, mirroring the deep SADCP trajectories. By contrast, during LDB, with weaker currents, the SedTrap Drifter moved in a similar fashion to the more shallow SADCP and SVP paths. The LDC telemetry, having been cut short, is not useful here. Perhaps hidden in the data is some linear combination of SADCP and SVP trajectories that can best explain why the SedTrap Drifter moved as it did. Although perhaps useful for OUTPACE, this would be fitting a model to a particular drifter configuration, and remove future

flexibility in drifter design. Hence, we chose not to pursue this calculation.

The different trajectories taken between the AQUADOPP, SADCP, and SVP demonstrate how trying to follow a Lagrangian perspective can get quickly complicated within a few days, if not sooner (Ohlmann et al., 2017). Even measurements at similar depths between different instruments do not correspond, such as between the AQUADOPP and SADCP at 55 and 52m, respectively, for LDB. Where they do correspond, such as for LDA, this may be due to a strong signal, which is also problematic

due to the shear involved. As mentioned in Sect. 3.3, the drift between AQUADOPP and SADCP during LDA and LDB were

comparable, despite the differing correlations, due to the fact that currents were stronger at LDA. Additionally, while both AQUADOPP and SADCP integrations started from a chosen initial position, the timeseries data reflect measurements taken on moving platforms, i.e. the SedTrap Drifter and the ship, respectively. All these complicating factors considered, the conservative route, employed here, is to consider each source of data and give it the best chance to refute the assumption that advection was weak enough that the spatial boundary determined by tracer analysis was not crossed. This procedure is the recommendation the authors provide subsequent to having done this analysis: while no individual trajectory will likely be the right one, exploring enough of the variability between them should give an idea of whether other water masses are present.

## 4.3 Biogeochemical Sampling Limitations

The protocol established in this study has identified the conditions in which a quasi-Lagrangian drift array can be considered successful in sampling one water mass, and ostensibly a single biogeochemical environment. The results from the three deployments during OUTPACE provide an emerging picture of what is exactly observed during the 3-5 days of sampling. The trajectories shown in Fig. 15 demonstrate that rather than a single, one-dimensional profile, vertical shear means that observations are of water from an extended area, reflecting a spatial sampling footprint. Sometimes water is sufficiently different that changes can be seen during the in situ timeseries, and these layers can be isolated for closer inspection. For the rest of the layers, the present method produces the $R_Z$ limit for the sampling footprint.

The existence of the sampling footprint requires biogeochemical investigators to re-consider a few assumptions. As stated in the Introduction, this analysis uses only physical, and not biological, data. Therefore, one must ask whether horizontal gradients exist at smaller scales. The only way to characterize this is to make measurements through a gradient, though this can be difficult with the need to produce measurements over a large area within a short, quasi-synoptic timescale. Since there exist at least some trajectories for each LD station that approached $R_Z$, the risk of encountering gradients is always present. Thus, for each biogeochemical measurement, the assumption of whether horizontal T-S gradients reflect biogeochemical gradients must be validated. For OUTPACE, use of a Moving Vessel Profiler (MVP) to survey an extended area around each LD station was planned, but technical difficulties prevented their use in the application of this method, though the data were useful in another study (de Verneil et al., 2017).

Furthermore, the current approach focuses on individual density layers. Measurements that reflect time- and depth-integrated processes, such as particle settlement in the sediment traps, require more assumptions before they can be considered to represent a single environment. For sediment traps, depending on sinking particle velocities, the 3-5 day timeframe may not be long enough to represent the entire water column, ie the sinking of surface particles to the sediment trap depth. Since near-surface trajectories for all three stations were able to get near $R_Z$ in 5 days, any process with longer timescales would probably produce measurements contaminated by a water mass change. Additionally, if water at different depths arrive from entirely different directions, gradients in any of these layers will influence the result. For example, particles sinking from the 1024.5-1025 $\mathrm{kg\,m^{-3}}$ density layer during the later part of LDC will influence subsequent sediment trap collection. While water mass changes near the sediment trap depths are unlikely due to weaker currents, the example of LDC suggests the probability is non-negligible. In our study, since the CTD casts were focused on the upper 200 m of the water column, there were insufficient

T-S measurements to conduct the timeseries analysis for all the sediment traps. We therefore cannot use this method to evaluate if the trap data reflect the same upper water column as observed during each LD station.

In summation, the inability of the quasi-Lagrangian drifter to follow the water in a sheared flow requires the investigator to consider the spatial scale of horizontal gradients, which might be depth-dependent, and then to ask whether the measurement they are making involves integrating over timescales and depths that mean the measurements being made do not reflect the other in situ measurements.

## 5 Conclusions

The methodology applied in analyzing the tracer data for OUTPACE is reflective of a few characteristics specific to this region and to biogeochemical datasets more generally. First, in an ideal situation, the T-S analysis would determine the component water masses. While this might be achievable for some well-studied regions of the world ocean, this is not generally applicable. Second, alternative well-defined water mass properties, such as dissolved oxygen, tend to be applicable only at depth, such as through the thermocline. Naturally, this is because photosynthesis, respiration, air-sea surface flux, and a panoply of other biogeochemical and physical processes occur near the surface and impact these tracers, making them not conservative. Unfortunately, we are precisely interested in analyzing the water properties of the surface ocean where these processes are most intense. As a result, methodologies need to be developed so that standard T-S measurements, unaffected by biogeochemical processes, can be used to quantify the effect of the physical circulation upon the biological environment encountered in a field campaign.

The multiple in situ data sources compiled during the LD stations of the OUTPACE cruise allowed for the determination of whether the ship, and its associated quasi-Lagrangian drifting array, sampled the same physical environment, which was divided into density layers. The procedures used to do this consists of several steps. First, one needs to look for large gradients (physical or biogeochemical) and structures of circulation (fronts or eddies) that would both impact the trajectory of the drifter and bring different water masses into close contact. Then, a statistical baseline is created from the T-S variability seen during the station's occupation and transformed into spice-density coordinates. Comparison of independent T-S data to the baseline is used to calculate Z-scores, first to analyze the in situ timeseries of the drifter deployment, and then to establish a spatial scale $R_Z$ beyond which the T-S differences amount to a change in the physical environment. While this scale is depth-dependent due to vertical shear, for the OUTPACE cruise a conservative $R_Z$ applied to all depths was found to be close to the Rossby radius $R_D$, in the 35-55 $\mathrm{km}$ range. The last step is to then use all available data regarding currents and drifter positions to evaluate whether any water parcel could have ostensibly traveled farther than $R_Z$. During OUTPACE, some density layers were shown to be variable enough to represent a change in water mass, and some trajectories surpassed $R_Z$. For the majority of measurements, however, this did not occur.

The methodology used in this study provides a framework wherein readily available T-S data can be used to answer the same question (whether a single physical environment was sampled following a quasi-Lagrangian drifting mooring) for other oceanographic cruises. More traditional methods, such as absolute water mass determination, or using alternative tracers such

as dissolved oxygen, require prior knowledge of a given region or are not applicable in the euphotic zone where biogeochemical measurements are made. While sampling in a Lagrangian manner is preferable to not attempting to follow a water parcel at all, the inevitable failure to be truly Lagrangian with these platforms should be recognized so that experiments are not allowed to either last too long or be deployed in an inappropriate flow regime. Regarding the use of this methodology, we give a few recommendations for future cruise sampling:

- Maximize use of remote sensing data during the cruise to identify possible mesoscale features to either avoid or sample inside of. This can be achieved with software such as SPASSO (Petrenko et al., 2017).

- Upon arrival at the selected site, a deep CTD cast below the thermocline can be used to quickly calculate the local $R_D$ Rossby radius in real time and produce a rough estimate for maximum spatial scale. In patchier environments, this scale might be too large, and must be reduced.

- Before and after each station, sample with a surveying instrument such as ISIIS, SWIMS, SeaSoar, or MVP beyond $R_D$ to get depth-resolved data at intermediate scales.

- If possible, mount CTDs and current meters on the quasi-Lagrangian drifting array (perhaps a sediment trap that does not need to be removed constantly). Multiple independent observations over a large range of spatial scales are essential to calculate robust $R_Z$ estimates.

- Deploy surface drifters to compensate for the lack of SADCP observations near the surface and to provide spatial context beyond the research vessel.

*Acknowledgements.* The authors would like to thank Gilles Rougier for his help in acquiring and processing CTD, SVP drifter, and SedTrap Drifter data. Marc Picheral is thanked for his help with sampling and processing CTD data. Olivier Desprez de Gésincourt is also thanked for use of and processing of AQUADOPP and Microcat CTD data. Rick Lumpkin and Shaun Dolk from NOAA/AOML are greatly thanked for providing and sharing SVP drifters and data. The authors are also indebted to the captain and crew of the *R/V L'Atalante* for their outstanding assistance and role in the OUTPACE cruise. This is a contribution of the OUTPACE (Oligotrophy to UlTra-oligotrophy PACific Experiment) project (Moutin and Bonnet, 2015) funded by the French national research agency (ANR-14-CE01-0007-01), the LEFE-CyBER program (CNRS-INSU), the GOPS program (IRD), and CNES (BC T23, ZBC 4500048836). The OUTPACE cruise was managed by MIO (OSU Institut Pytheas, AMU) from Marseille (France). Catherine Schmechtig is thanked for LEFE-CyBER database management. Satellite SST, chl *a*, and altimetry data have been provided by CLS in the framework of CNES funding. Aurelia Lozingot is acknowledged for administrative aid for the OUTPACE project.

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

**Table 1.** Start and stop times for the timeseries data in each LD station of OUTPACE. Times expressed in DD/MM/YYYY HH:MM:SS (GMT) format. When multiple instruments were deployed or multiple discrete observations made, this is also noted.

| STATION | | | | |
|---|---|---|---|---|
| **LDA** | **Instrument** | **CTD Rosette** | **SVP** | **AQUADOPP** |
| Longitude: | # | 46 casts | 3 drifters | 6 deployed |
| 123.34° E | Start | 2/25/2015 14:09:18 | 2/25/2015 20:00:00 | 2/26/2015 22:40:00 |
| Latitude: | Stop | 3/02/2015 16:10:10 | 3/02/2015 22:00:00 | 3/02/2015 16:10:00 |
| 18.32° S | Duration | 5d 2h 0m 52s | 5d 2h | 3d 17h 30m |
| | **Instrument** | **SADCP 150** | **SADCP 38** | **SedTrap Position** |
| | Start | 2/25/2015 14:09:57 | 2/25/2015 14:10:26 | 2/25/2015 19:01:13 |
| | Stop | 3/02/2015 16:09:51 | 3/02/2015 16:08:38 | 3/02/2015 22:00:00 |
| | Duration | 5d 1h 59m 54s | 5d 1h 58m 12s | 5d 2h 58m 47s |
| **LDB** | **Instrument** | **CTD Rosette** | **SVP** | **AQUADOPP** |
| Longitude: | # | 47 casts | 6 drifters | 6 deployed |
| 123.34° E | Start | 3/15/2015 12:04:44 | 3/15/2015 10:00:00 | 3/15/2015 23:10:00 |
| Latitude: | Stop | 3/20/2015 14:16:13 | 3/20/2015 23:00:00 | 3/20/2015 14:15:00 |
| 18.32° S | Duration | 5d 2h 11m 29s | 5d 13h | 4d 15h 5m |
| | **Instrument** | **SADCP 150** | **SADCP 38** | **SedTrap Position** |
| | Start | 3/16/2015 08:51:53 | 3/15/2015 23:06:31 | 3/15/2015 12:15:48 |
| | Stop | 3/20/2015 14:15:54 | 3/20/2015 14:14:50 | 3/20/2015 21:00:00 |
| | Duration | 4d 5h 24m 1s | 4d 15h 8m 19s | 5d 8h 44m 12s |
| **LDC** | **Instrument** | **CTD Rosette** | **SVP** | **AQUADOPP** |
| Longitude: | # | 46 casts | 4 drifters | 6 deployed |
| 123.34° E | Start | 3/23/2015 23:10:57 | 3/23/2015 12:00:00 | 3/23/2015 23:25:00 |
| Latitude: | Stop | 3/28/2015 14:32:30 | 3/28/2015 22:00:00 | 3/28/2015 14:30:00 |
| 18.32° S | Duration | 4d 15h 21m 33s | 5d 10h | 4d 15h 5m |
| | **Instrument** | **SADCP 150** | **SADCP 38** | **SedTrap Position** |
| | Start | 3/23/2015 12:08:12 | 3/23/2015 12:06:55 | 3/23/2015 12:19:55 |
| | Stop | 3/28/2015 14:30:31 | 3/28/2015 14:31:39 | 3/26/2015 03:31:34 |
| | Duration | 5d 2h 22m 25s | 5d 2h 24m 44s | 2d 15h 11m 39s |

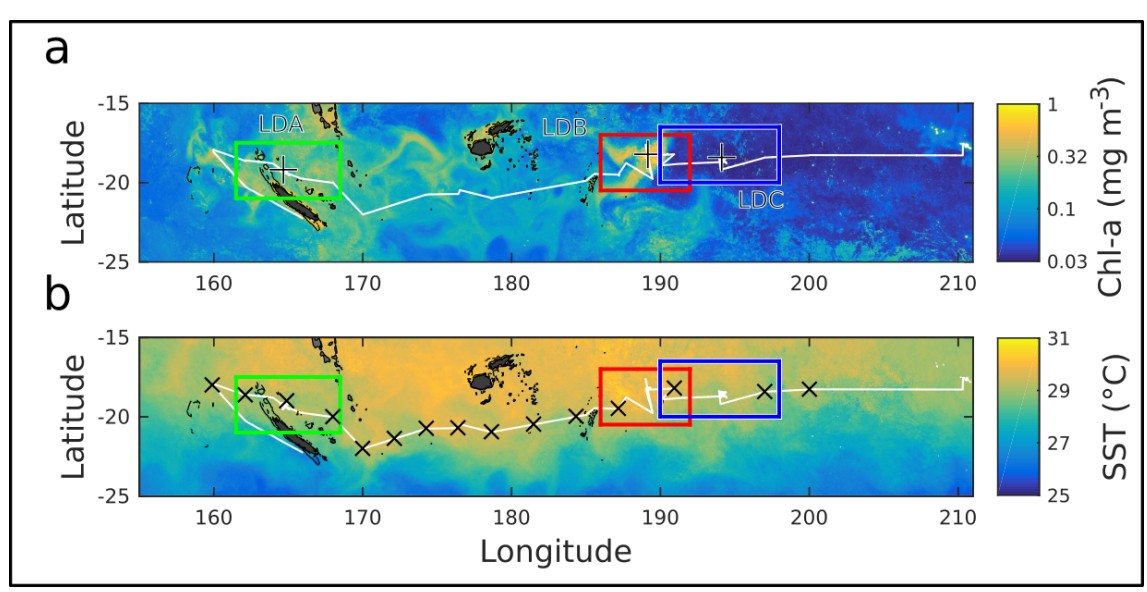

**Figure 1.** Satellite surface (a) chl-*a* and (b) SST for the OUTPACE cruise. Pixel data are weighted by the normalized inverse distance squared between each pixel and the RV *L'Atalante*'s daily position over the 42 days of OUTPACE. Shiptrack shown in white. LD station locations shown with black +'s. Domains used in Fig. 2 are shown by color-coded rectangles, with green for LDA, red for LDB, and blue for LDC.

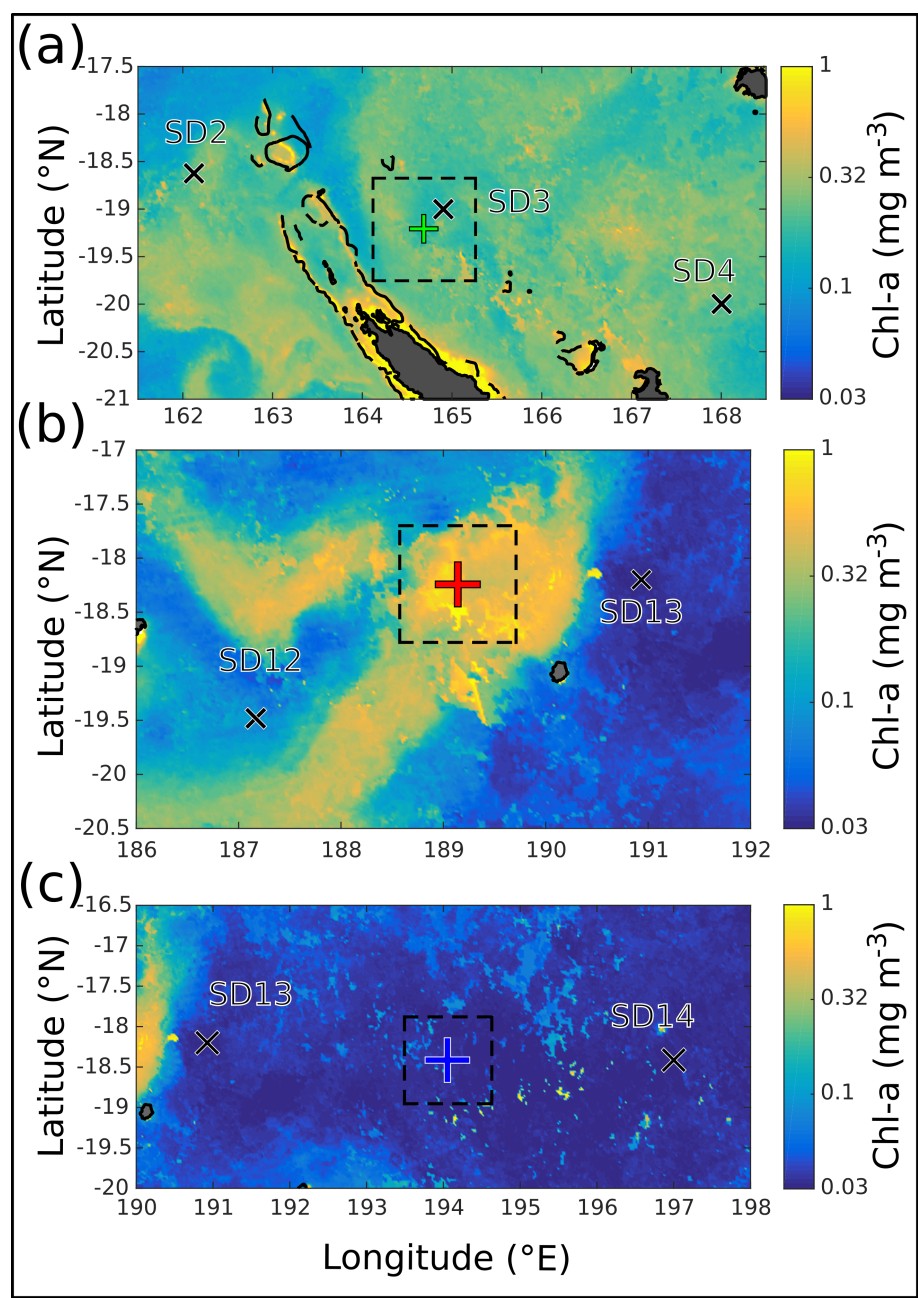

**Figure 2.** Satellite chl-*a* around (a) LDA, (b) LDB, and (c) LDC. SD stations shown by black x's, land is shaded gray, and coastlines/reefs plotted in black. LD stations shown by +'s following the color code from Fig. 1. Squares with 120 km to a side plotted around each LD station to represent approximate Rossby radius $R_D$.

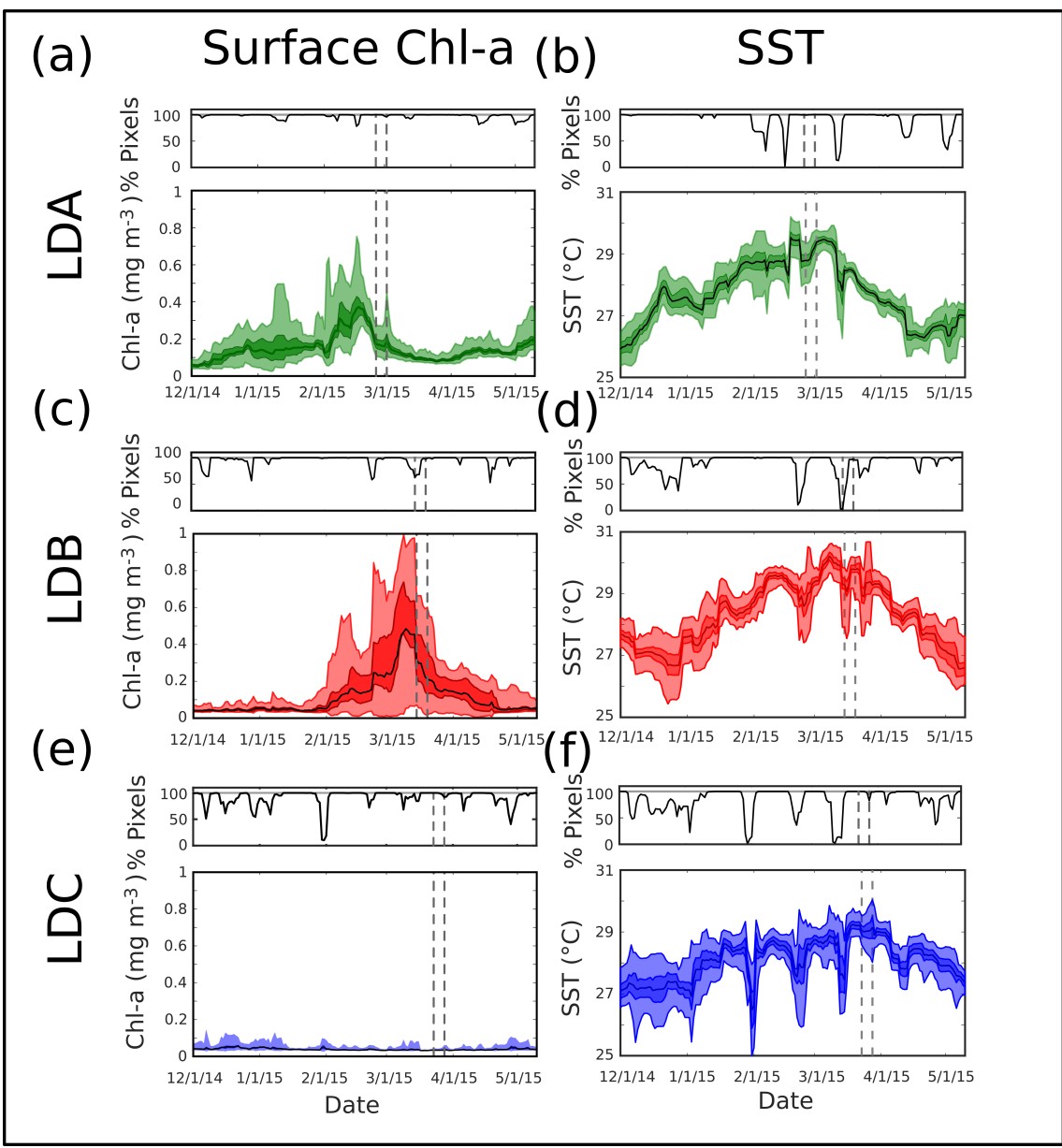

**Figure 3.** Timeseries of surface chl-*a* and SST, respectively, for (a,b) LDA, (c,d) LDB, and (e,f) LDC. Intervals of LD sampling shown with gray dashed lines. Mean values are plotted in black, with darker shades representing the 25-75% interval and lighter shades for 1-99%. Subpanels above each timeseries depict the % of pixels with data. All data come from within the 120 km × 120 km squares shown in Fig. 2.

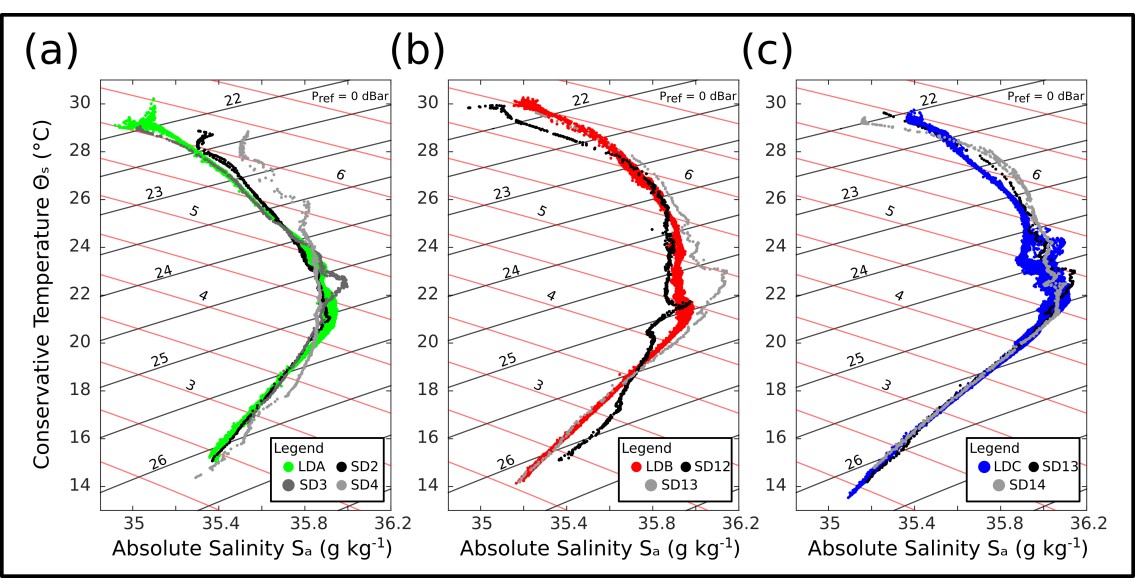

**Figure 4.** T-S diagrams of (a) LDA, (b) LDB, and (c) LDC and surrounding stations. LD stations are color-coded, and SD stations different shades of gray. Isopycnals are displayed in black, with isopleths of spice shown in red.

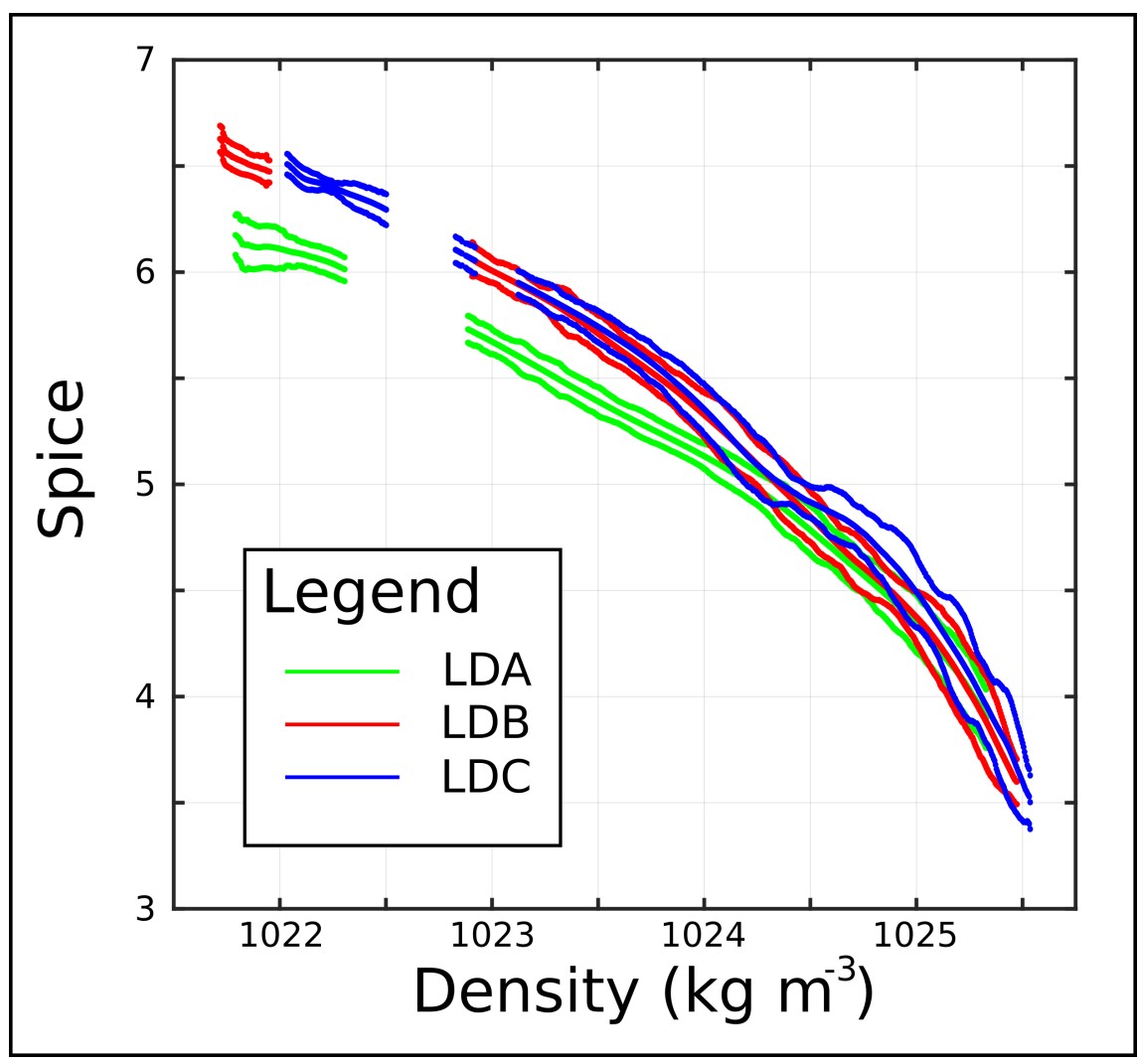

**Figure 5.** Statistical LD baseline of spice versus potential density for (a) LDA, (b) LDC, and (c) LDC. Mean values plotted in between envelope of $\pm 2\ SErr_{obs}$.

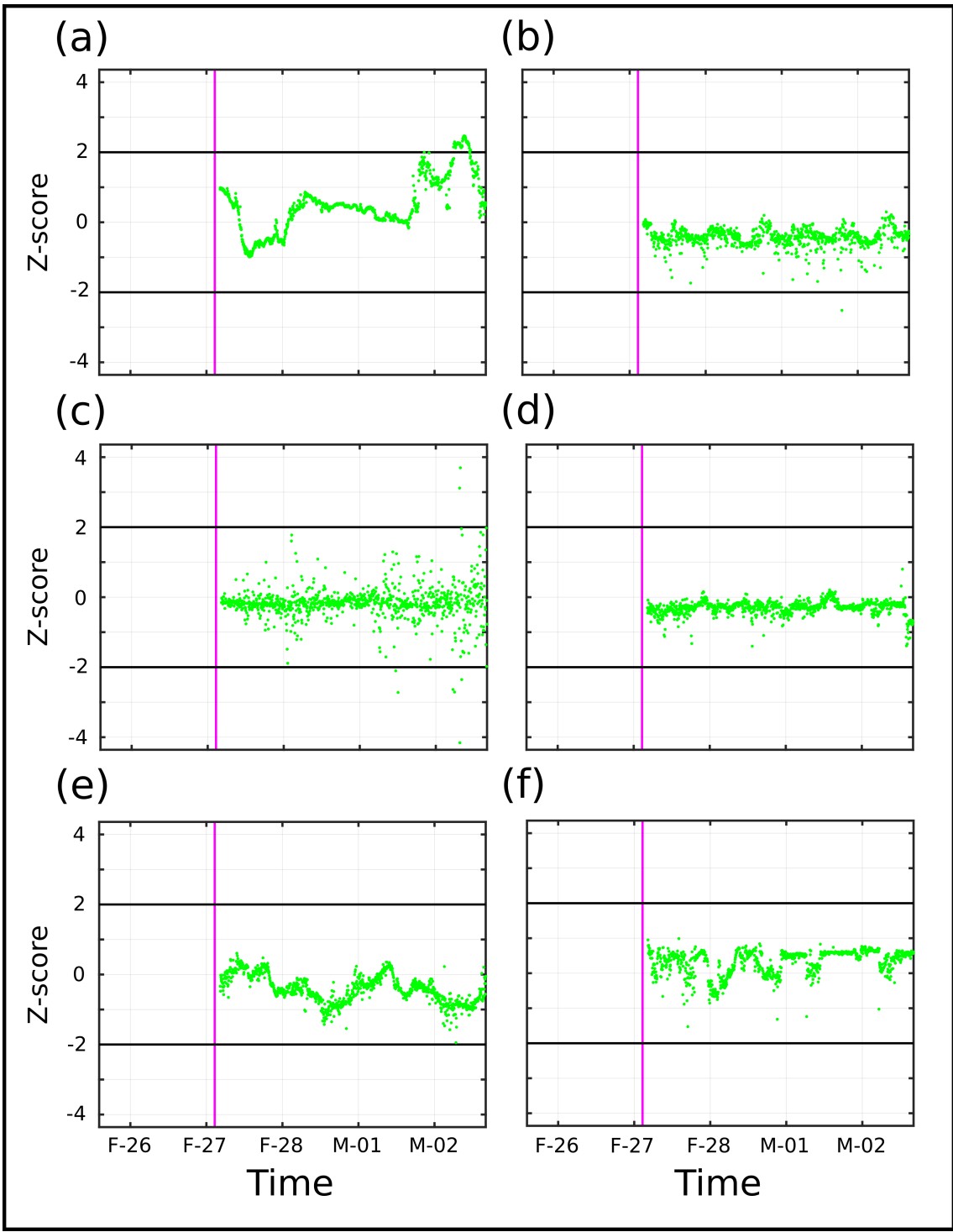

**Figure 6.** SedTrap drifter Z-score timeseries for (a) 14, (b) 55, (c) 88, (d) 105, (e) 137, and (f) 197 m depth. End of inertial period timeframe for baseline definition plotted in magenta. Z=-2 and 2 plotted in black.

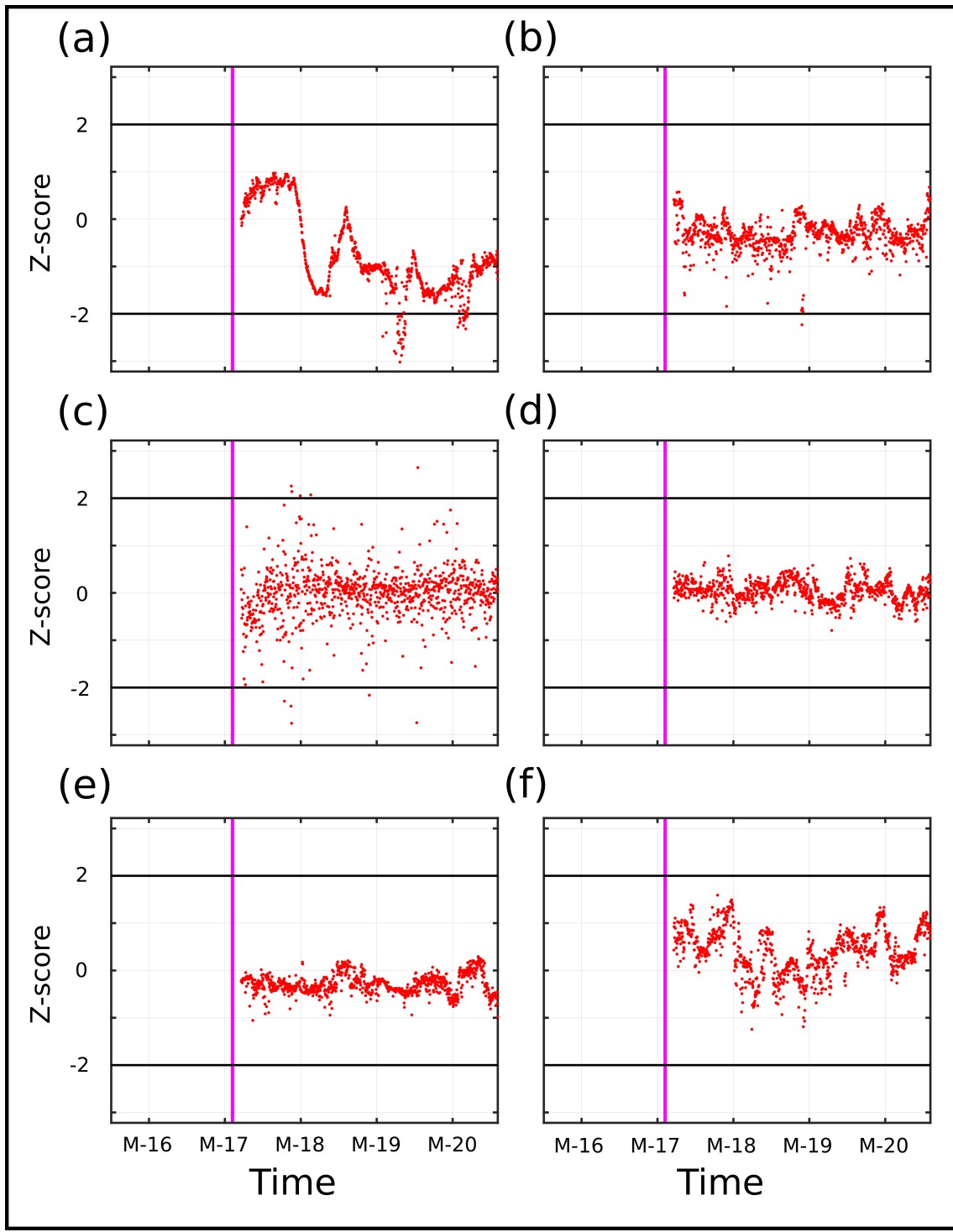

**Figure 7.** Same as Fig. 6 but for LDB.

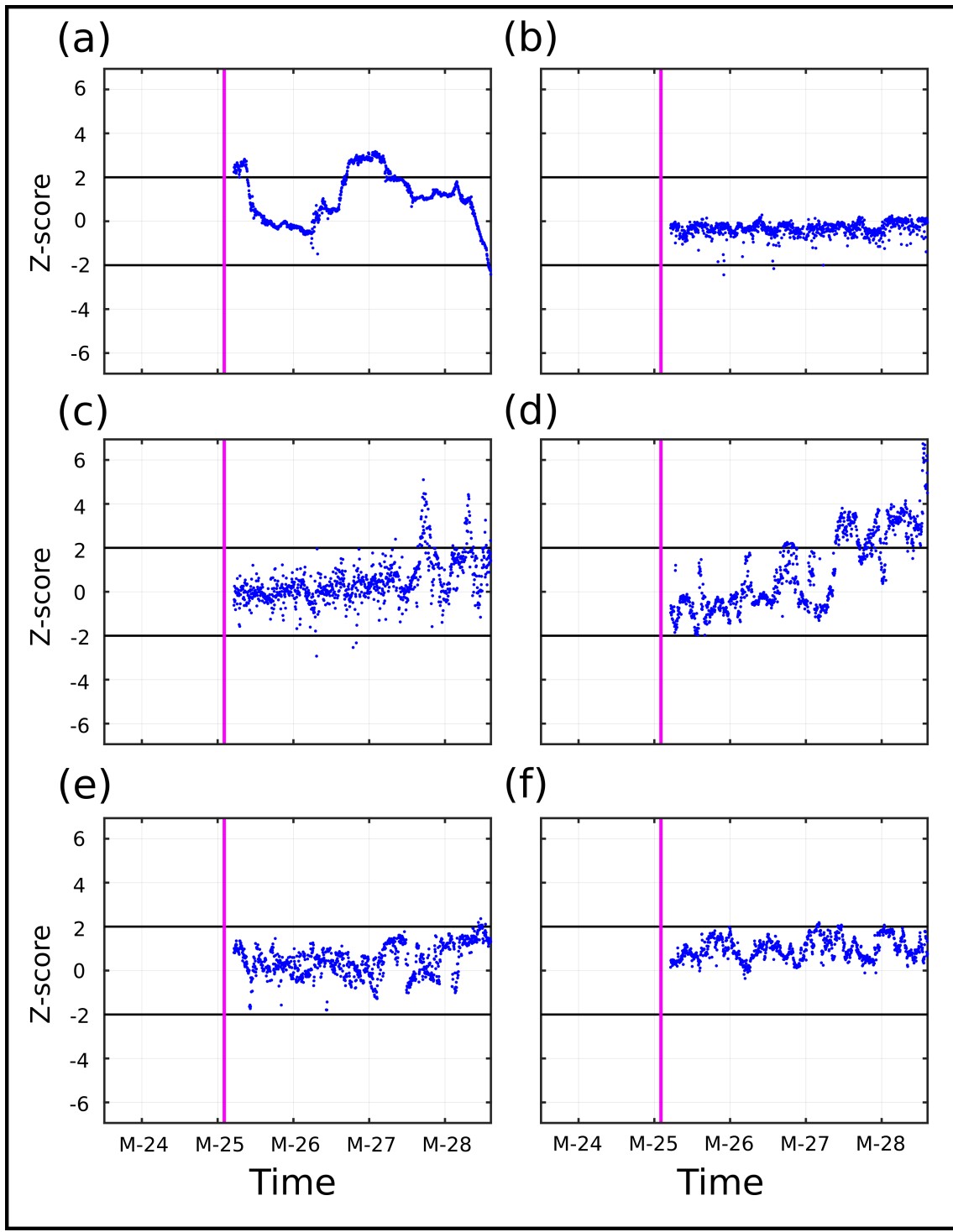

**Figure 8.** Same as Fig. 6 but for LDC.

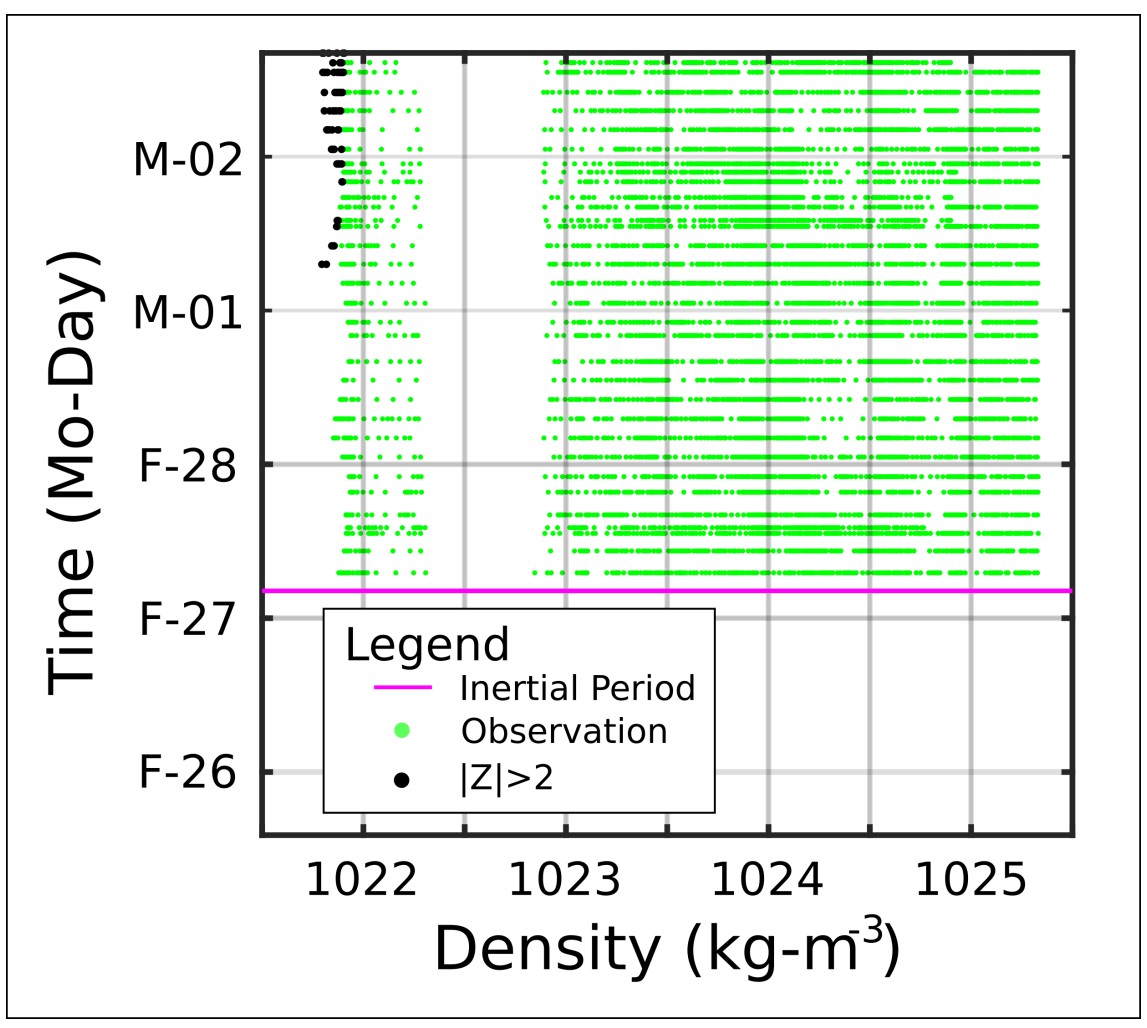

**Figure 9.** Timeseries of CTD observations for LDA. Observations with |Z|<2 plotted in green, |Z|>2 in black. End of statistical baseline definition period shown in magenta.

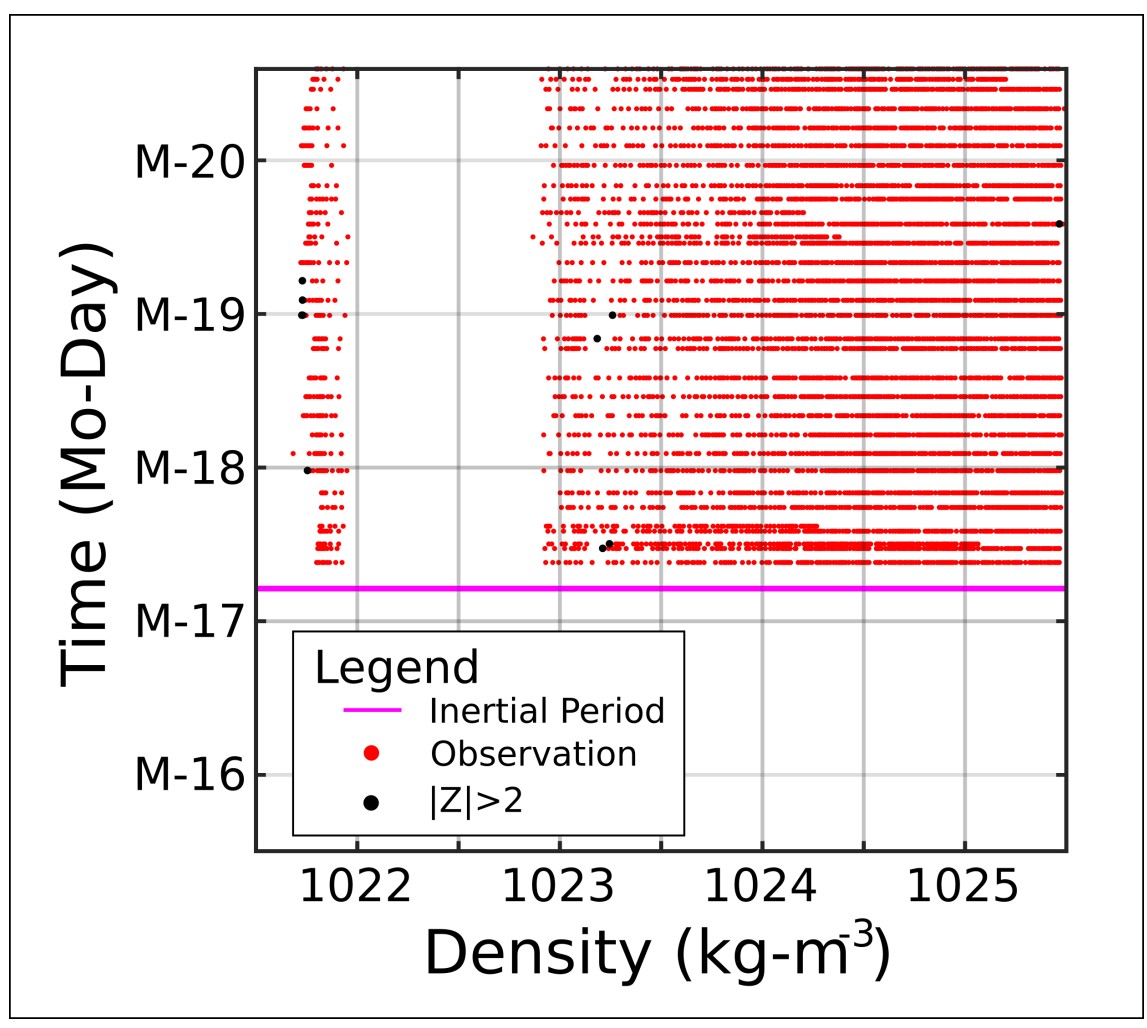

**Figure 10.** Same as Fig. 9 for LDB, with |Z|<2 observations plotted in red.

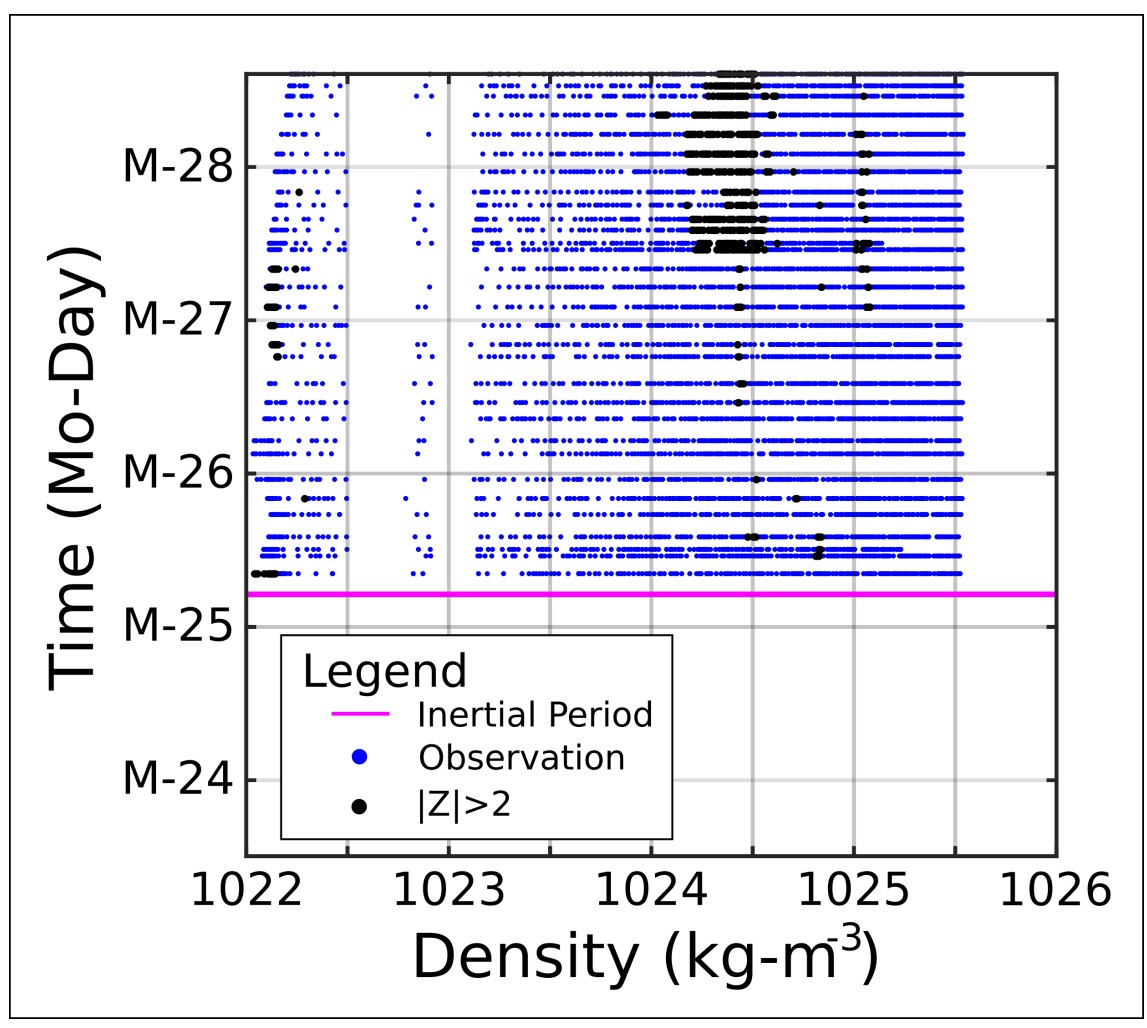

**Figure 11.** Same as Fig. 9 for LDC, with |Z|<2 observations plotted in blue.

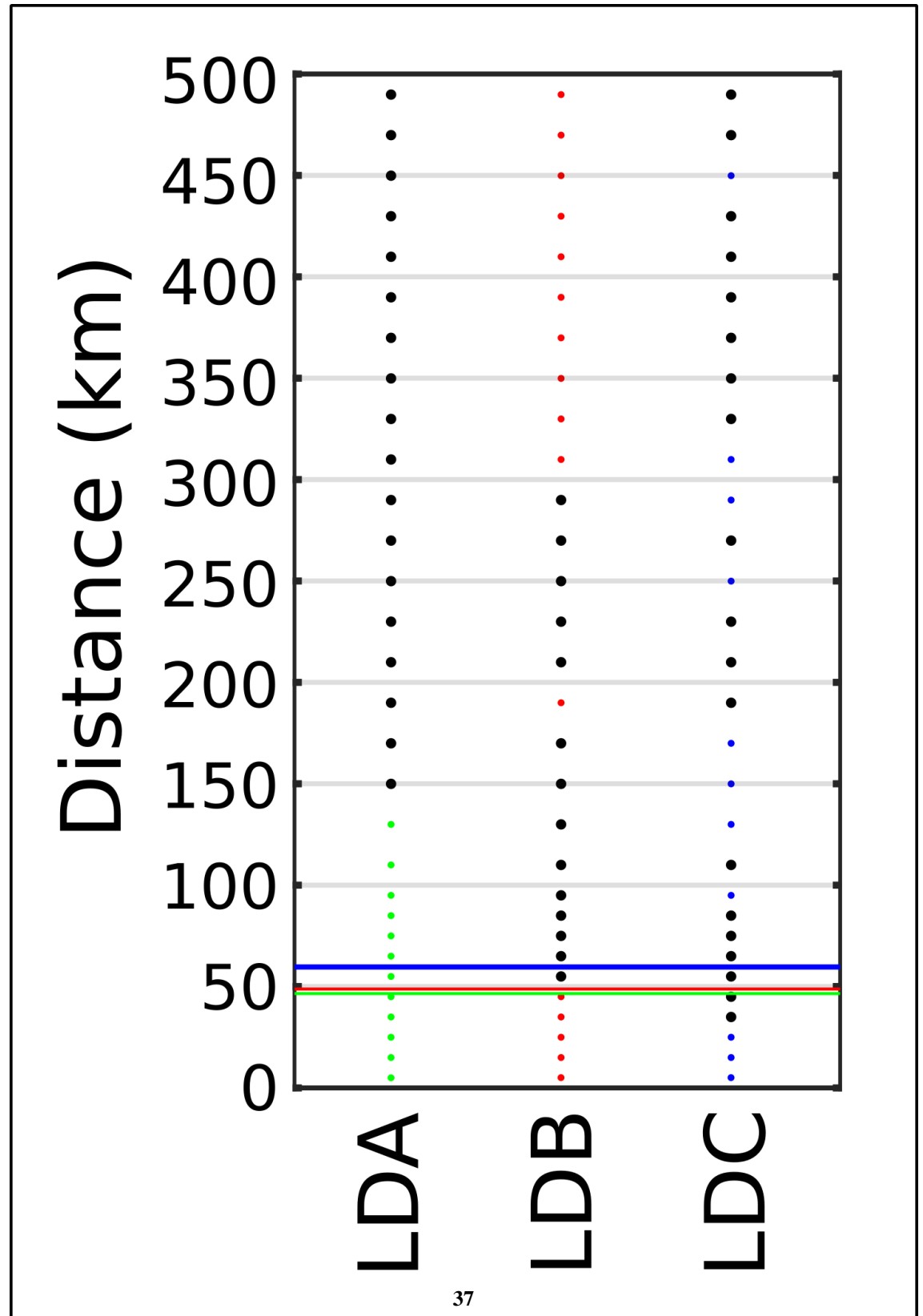

**Figure 12.** TSG Z-score over distance for LDA( left, green), LDB (center, red), and LDC (right, blue). |Z|>2 shaded black. Rossby radii $R_D$

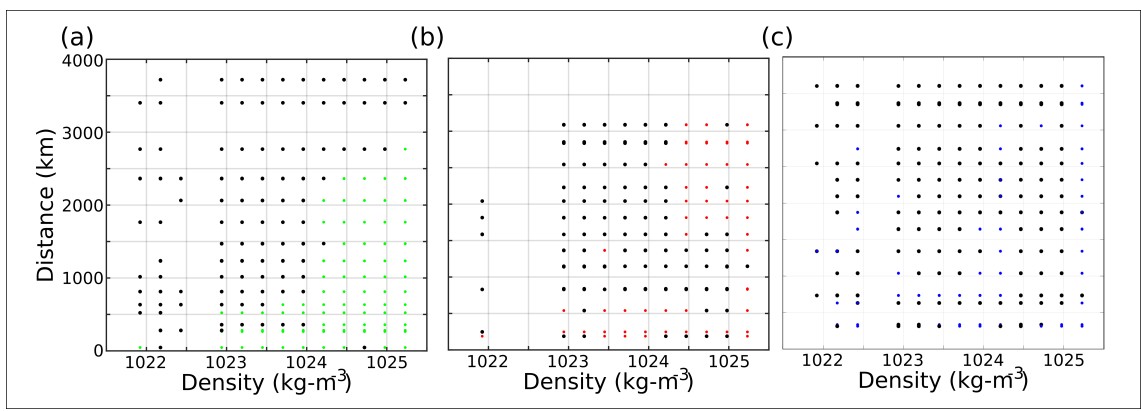

**Figure 13.** SD station Z-score over distance for (a) LDA, (b) LDB, and (c) LDC. |Z|>2 shaded black. Rossby radii $R_D$ distance plotted in horizontal dashed lines, color-coded to the LD stations.

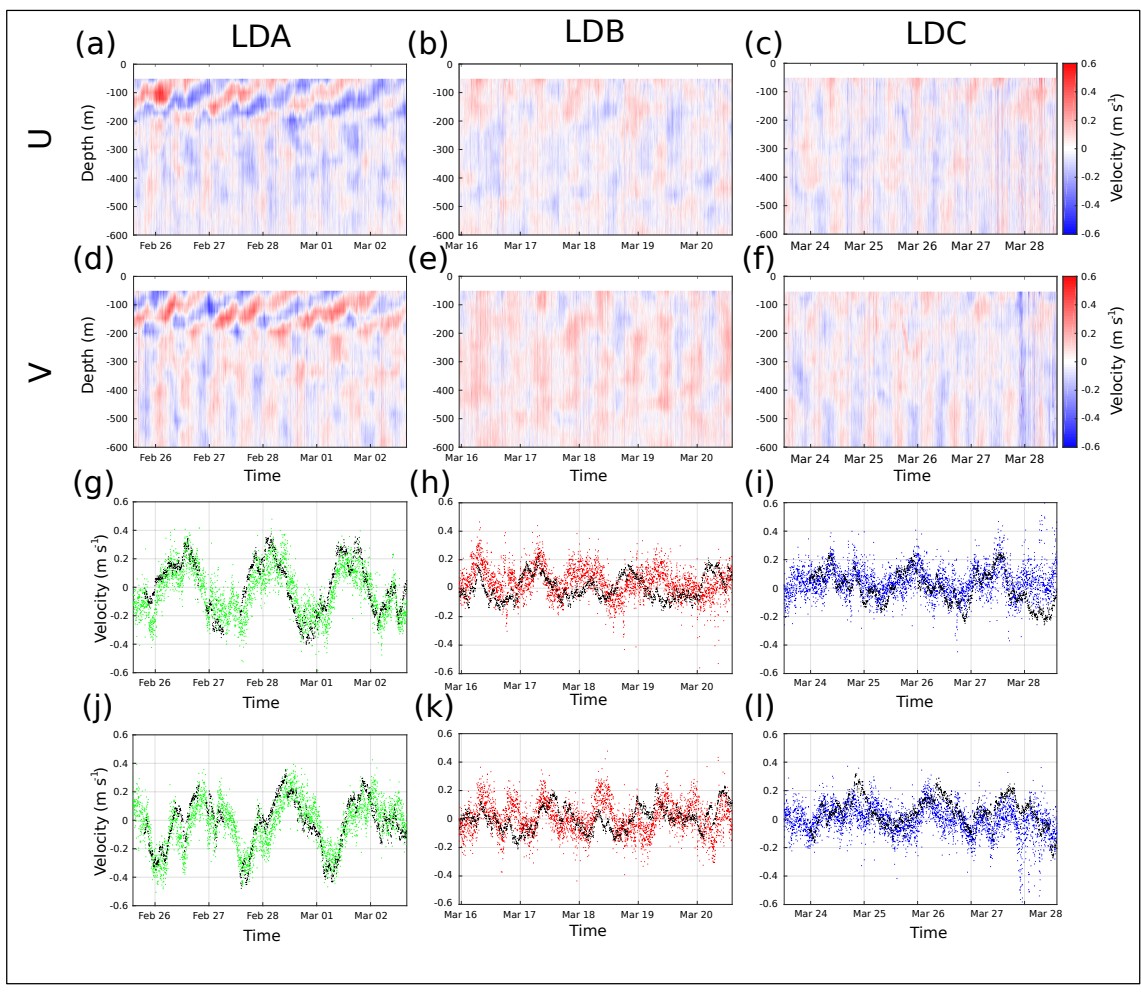

**Figure 14.** Timeseries of 38 kHz SADCP u and v components over depth for, respectively, (a,d) LDA, (b,e) LDB, and (c,f) LDC. Comparison of 55 m Aquadopp u and v components with 52 m SADCP u and v for, respectively, (g,j) LDA, (h,k) LDB, and (i,l) LDC. Aquadopp measurements shown in black, SADCP measurements follow the color code. Units are in $\mathrm{ms}^{-1}$.

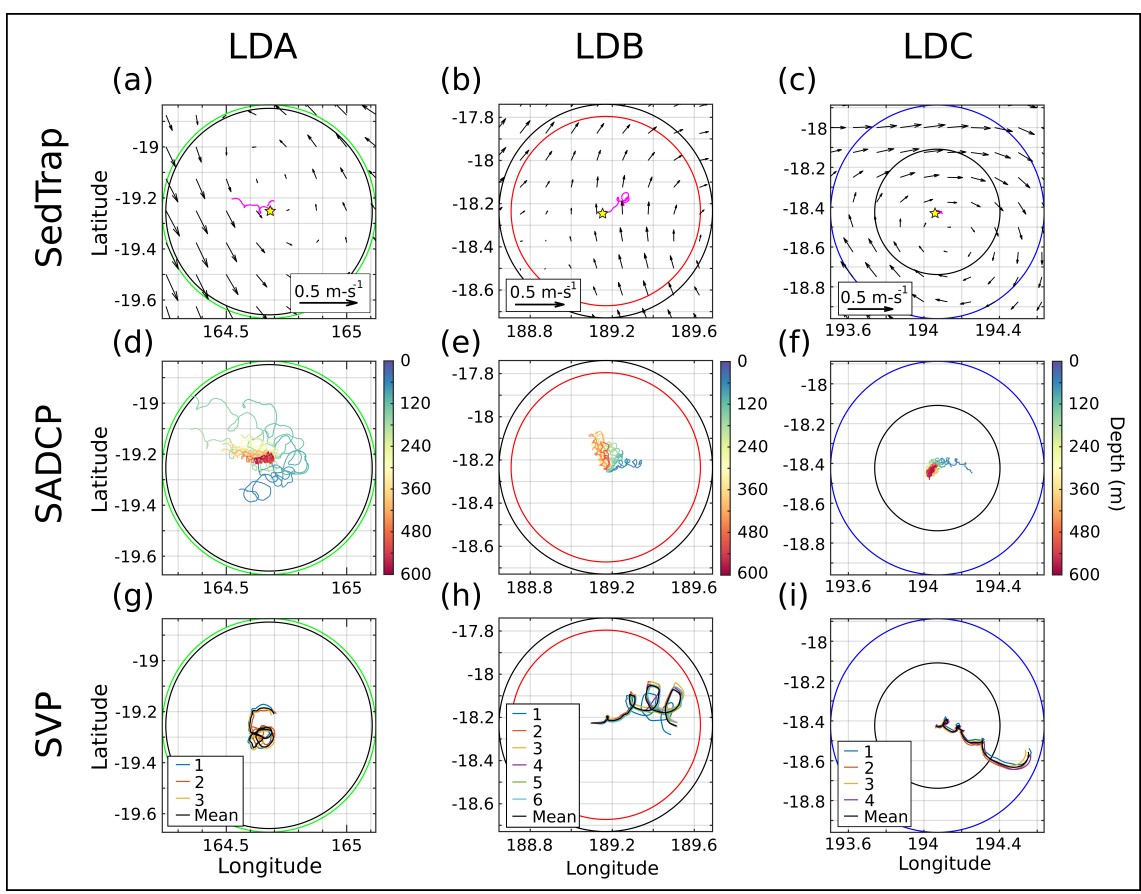

**Figure 15.** Observed and calculated trajectories for currents analysis. Data from LDA, LDB, and LDC, shown in left, center, and right columns, respectively. Top row (a-c): Observed trajectory of SedTrap drifter plotted in magenta. Time-averaged altimetry-derived surface currents shown with black arrows. Rossby radius RD traced as a color-coded circle, and RZ, the calculated spatial scale, in black. Starting position of SedTrap drifter shown with a star. Middle row (d-f): Calculated SADCP 38 kHz trajectories of water at each depth down to 600 m. Bottom row (g-i): Observed SVP drifter trajectories, with mean trajectory plotted in black.

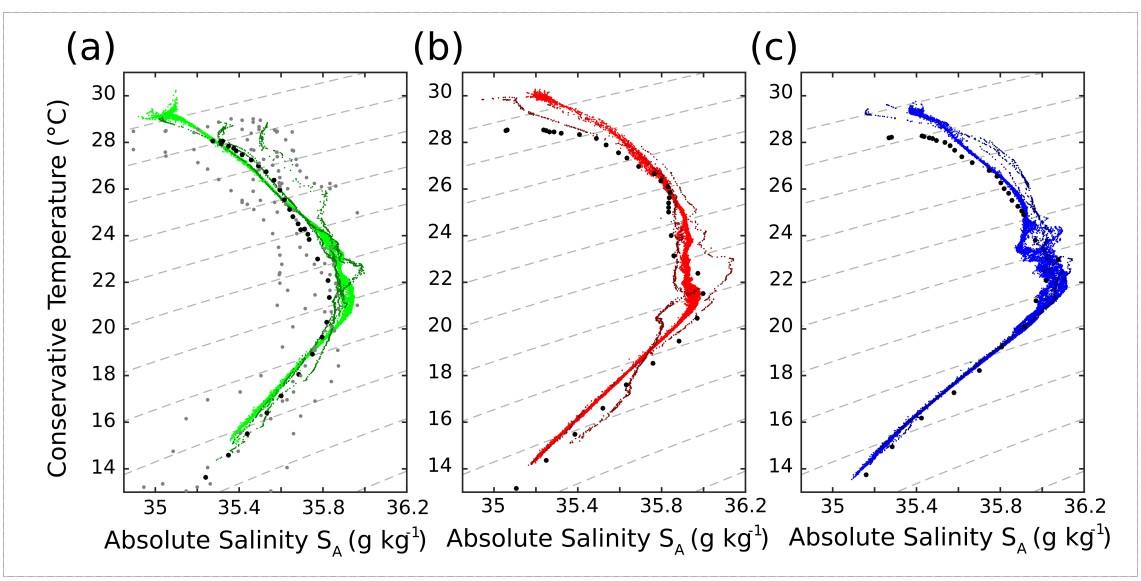

**Figure 16.** Supplementary Figure 1. T-S Diagrams of LD stations and neighboring SD stations, as in Fig. 4, with World Ocean Atlas 2013 v2 climatological profiles at the LD station positions superimposed. Climatological monthly means shown in black, with extrema of T-S shown in gray, as calculated by +/- 2 standard errors. Insufficient data at LDB and LDC preclude calculating these errors.

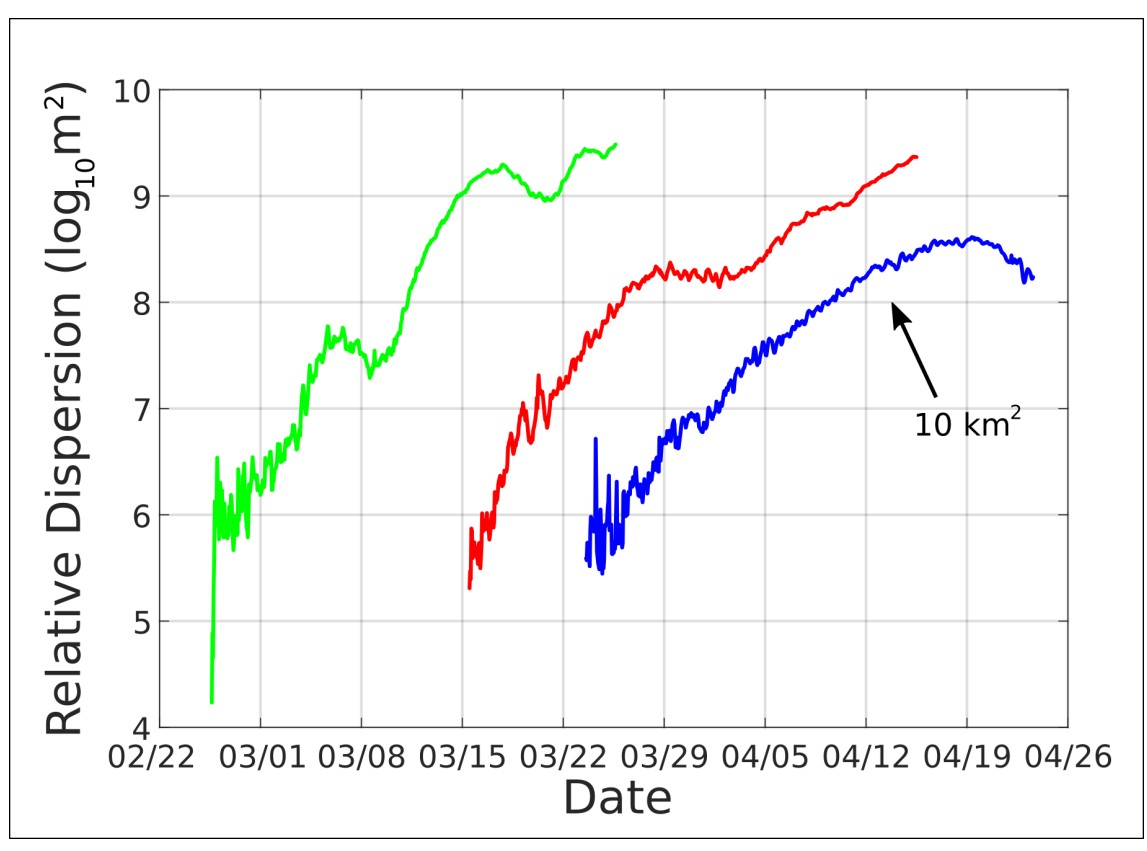

**Figure 17.** Supplementary Figure 2. Timeseries of SVP drifter relative dispersion during LD stations. Sampling start and end times demarcated by dashed gray lines. Data for the first month after SVP deployments are shown.