# Peer review of "OUTPACE long duration stations: physical variability, context of biogeochemical sampling, and evaluation of sampling strategy"

_Biogeosciences, 2017_

## Referee Comment (RC1) · Anonymous Referee #1 · 2 Jan 2018

General comments This is a quality paper, suitable for publication, sharply focussed on specific issue of whether a "Lagrangian" biogeochemical drift station lasting a few days is truly following the same piece of water. The methodology is sound, but focused only on the tools and environment associated with the OUTPACE program. Other environments, such as the subpolar North Atlantic during the Spring Bloom would have very much smaller scales.

The major assumption of the analysis is that temperature and salinity variability can be used as a proxy for biogeochemical variability. Clearly if there is a strong physical change, then it is more likely that they will also be different biogeochemical regimes.

However, there can also be biogeochemical variability that is unrelated to physical variability, even variability in biogeochemical variables purely due to physical affection that is not apparent in the temperature and salinity fields. Thus the analysis presented here is closer to a necessary condition for a good Lagrangian trajectory, but is not necessarily a sufficient one.

The only true measure of biogeochemical variability around a quasi-Lagrangian trajectory is the actual variability. This will be different depending on the variable measured and the depth at which it is measured. Given the ability to make simultaneous measurements of many biogeochemical quantities from multiple drifting platforms, and the ability to autonomously survey around a platform, measuring in many places around a Lagrangian platform is becoming increasingly possible.

Specific comments Section 2.1 - How many drifters were deployed that were different from the drifting moorings? Where were they deployed relative to the moorings?

How were the SD stations placed relative to the LD stations? Was there any attempt to do a survey around the LB stations?

Figure S3 - Would be nice to additionally label the dispersion plot with units of meters and kilometers to guide the reader, i.e. $10^8$ m$^2$ is (10 km )$^2$

Recommendations - Although the first Rossby radius appears to be a good scale here, it is certainly much too large in other, more patchy environments.

---

## Referee Comment (RC2) · Anonymous Referee #2 · 3 Jan 2018

The paper assesses the Lagrangian nature of drifting sediment traps using physical data collected on a drifters and from ship observations of the surrounding waters. While the motivation of the study is sound I found the presentation difficult to follow. If the intention is to both assess the Lagrangian nature of the OUTPACE deployments and provide a generic method for such an assessment then the presentation of the method needs to be improved to enable the reader to clearly follow the approach. I consider the paper requires major revisions to address this point and hence be acceptable for publication. In revising the paper the authors need to address:

1. The way the method section is written it is not clear that you are assessing the

Lagrangian nature of the SedTrap drifter and how you use the collected data to do this. I like the use of Spice in the analysis but how is the baseline data profile determine - first hour, day ? of the SedTrap measurements? something else? Need to clearly define the data that went into the baseline definition before I can assess the validity of the method. Sounds like you used the CTD data but I would think you should use the initial profile of the SedTrap drifter and then look at changes in the water properties of the SedTrap Drifter to determine whether the float is Lagrangian.

2. The second step to the method is also not clear "evaluation of whether that scale was surpassed by the SedTrap Drifter ... ". What does this mean, why is it useful, how is it determined, and why does it differ from looking at the changes in the water properties of the SedTrap Drifter? Please expand this discussion so I can follow how the current data collected on the different platforms are used to assess the Lagrangian nature of the SedTrap drifters. The trajectories from the different velocity datasets appear significantly different and not consistent with the drift of SedTraps, what does this mean?

3. The method section also provides much additional information that does not help clarify what you are doing. What is the value in comparing to climatology for the Lagrangian assessment? Why discuss water mass breakdown if it is not used? Why even show the remotely sensed maps and evolution if they are not used to assess the Lagrangian nature of the SedTrap Drifters?

4. Is the Lagrangian nature satisfied for all depths in the SedTrap drifter deployment? One may expected a surface drifter may not represent the flow at depth. Please discuss.

5. The SedTrap drifters all drift less than 5.6 km - hard to think it is not Lagrangian since it barely moved. Why do analysis out to 1000 km when the drifter barely moves? Need to provide some context as to why you extend the analysis to much larger scale.

Some specific comments.

Abstract. l1-5 - not clear what a quasi-Lagrangian drifter approach is - deploy a drifter and follow it. Add a sentence to explain what this is.

pg3 line 23 - state how close the production line was to the SedTrap drifter?

pg4. How is the remote sensing data used to assess the Lagrangian nature of the SedTrap drifters?

- where do you use Mixed Layer depth in your analysis?

pg5 line 8 - to make it easier to follow say you are assessing the SedTrap Drifters since it is confusing when you use the quasi Lagrangian drifting mooring.

line 14 -how is comparing to climatology useful for you analysis? line 18-25 - again interesting information but not relevant here. Could go in the introduction as a way of testing Lagrangian nature of the sampled water. It is not part of your method.

pg 6 line 14-25 - Z scores are functions of density do they vary significantly with density line 34 -does the SedTrap Drifter have greater variability than TSG

pg8 why is the satellite data needed?

Table 2 variation in Dist(z=2) is huge what does this mean? It implies the calculations are very dependent on the data source? or you have not used the data appropriately

Figure 1. state it is a weighting of the 42 days of data based on inverse distance squared from the ship track. What remotely sensed data is used and what is it resolution in time and space?

Figure 2. Is there weighting of the remotely sensed data? If you need to show remotely sensed data I would use figure 2 only.

Figure 3. why show this plot? it is not a Lagrangian view of the data. It simply shows the variability in the region around the time of drifter deployment. How do you use it in your assessment?

Figure 4. Spice is not orthogonal to density - why not?

Figure 6. How do you measure distance for SedTrap drifter? How is distanced defined for the other data? Why go out to 1000 kms when the acceptable distance is less than 100 km?

Figure 7. Why are there such large differences between velocities in h, i, k, l panels?

Figure 8. rewrite caption to clarify what you are showing. What are the 3 rows showing? The trajectories appear quite different between the rows, why? and what implications does this have for the Lagrangian assessment?

---

## Short Comment (SC1) · 3 Jan 2018

This manuscript is concerned with the evaluation of the Lagrangian nature of series of measurements made using a drifting mooring as a reference center. Specifically, the main goal of the study is to define an objective measure to quantify the degree to which sampling along the trajectory of a drifting mooring can be considered Lagrangian (with the ability to choose a threshold value for that measure beyond which the observations will not be considered Lagrangian). This is a methodological topic, yet it should be of interest to a relatively broad audience concerned with physical and biogeochemical ocean observations. The text, figures and captions are clear and informative (except

in very few places, see below). On the other hand, the scientific context and working hypotheses underlying the study are not adequately posed in my opinion. Suggestions are provided below on how to improve this, which may simply involve modifications of the text. Alternatively, a more in-depth revision could make use of numerical simulations to carefully evaluate the proposed Lagrangian evaluation method.

Main comment: the scientific context and working hypotheses underlying the study are not adequately posed.

This is perhaps due to the fact that the authors are venturing into "new territory". There have been past studies attempting to quantify departures from pure Lagrangian motion associated with drifter trajectories (D'Asaro 2003) but the context of the present study differs. Indeed, the drifting moorings deployed during OUTPACE are not expected to precisely follow a water parcel of infinitesimal size but rather to remain in a given environment having a finite vertical heigh (defined in terms of density range in that study). The reader attention should be clearly drawn to this specificity at the beginning of the paper. An important consequence noticed by the authors (but in their final sections) is that, except in a perfectly barotropic flow, the mooring trajectory will not be truly Lagrangian at any vertical level. In this context, and ignoring vertical motion, the three key elements of the problem are 1) the vertical shear of the flow 2) the structure of the mooring and the vertical distribution of the drag force 3) the horizontal scales over which the environment can be considered homogeneous (the scale may vary as a function of depth, e.g., scales may be shorter near the surface where submesoscale turbulence tends to be intensified).

Several remarks are in order. With respect to 3) it seems difficult, except in rare cases, to define an average scale as done in the present study. Indeed the $R_z$ defined by the authors ignores the Lagrangian aspect of the circulation. A fast moving environment eg in the vicinity of a hyperbolic point, will be systematically poorly evaluated from the perspective of the ability of OUTPACE moorings to remain Lagrangian, but for erroneous reasons because only the intensity of the shear (which may or may not be stronger in

such regions) matters. Likewise, a mooring embedded into a mesoscale structure may lead to a near-perfect "Lagrangian trajectory" (in the sense that the mooring perfectly tracks at all depths the coherent water masses trapped into the eddy) while travelling distances greater than Rd (eg, on time scales of weeks to months). I understand that the OUTPACE setting is one in which mesoscale are weak and this is explicitly stated by the authors. But then, could the authors strive to precisely describe the kind of dynamical regime they are in, and how it affects the general problem that is the motivation of the study ? If mesoscale turbulence is irrelevant then waves with different scales should dominate, typically Rossby waves, inertial waves, and tides. And only the motions with time scales longer than the station duration impact the "quasi-Lagrangian strategy" through their vertical shear. This is explicitly mentioned by the authors (p15 lines 23-25) but again, these concluding remarks come way too late and should have guided the whole validation design.

In any event, whether or not the presented work is only valid in one regime or not, it is unlikely that a proper metrics for how Lagrangian the observations are can exclude the measured vertical shear in the area. As for the rule of thumb that displacements longer than Rd should raise suspicion, I would certainly argue that this is too general a statement to be supported by the present work. Likewise the suggestion that there might be inappropriate flow regimes (p17 line 15) is not well supported and may have to be reconsidered.

Overall, it seems to me that the introduction should i) present the processes that can break the Lagrangian nature of the sampling approach implemented by the authors ii) briefly review how different dynamical regimes may be differently affected 3) present the OUTPACE regime of interest in which they will develop their methodological approach. This would allow the authors to put their work on much firmer ground and help them write a more robust discussion section.

If the authors intend to keep using this type of observational approach, I would additionally suggest a numerical investigation in which drifter trajectories advected with

velocities computed by several weighted vertical averages are compared with trajectories for drifters that are localised at particular depths and thus Lagrangian (if w can be ignored). Such a study might well reveal the broad range of Rz spatial scales coexisting in a given location even with limited mesoscale activity. A numerical investigation with biogeochemistry would even allow the authors to explore the limitations raised by reviewer 1.

Minor comments: abstract: "homogeneous" may be better suited than "self-similar". p4 line 10: "Square regions ...". I find this sentence unclear. p9 line 22: Not clear. Rephrase perhaps as: "The increase in salinity maximum from LDA to LDC reflects ..." p15 line 33: " Rather than an elevated shearing apart ...". Awkward sentence

Reference: D'Asaro, E. A. (2003). Performance of autonomous Lagrangian floats. Journal of Atmospheric and Oceanic Technology, 20(6), 896-911.

---

## Author Comment (AC1) · 19 Feb 2018

Response to Review # 1 for OUTPACE LD Station Paper:

We thank Reviewer #1 for taking the time to read the submitted manuscript and providing their thoughts and comments. Below is the original response from Reviewer 1 (in Italics), with our own responses interspersed within. Manuscript changes are shown with additions in blue, deletions in red strikethrough.

Reviewer:
*General comments: This is a quality paper, suitable for publication, sharply focused on the specific issue of whether a "Lagrangian" biogeochemical drift station lasting a few days is truly following the same piece of water. The methodology is sound, but focused only on the tools and environment associated with the OUTPACE program. Other environments, such as the subpolar North Atlantic during the Spring Bloom would have very much smaller scales.*

*The major assumption of the analysis is that the temperature and salinity variability can be used as a proxy for biogeochemical variability. Clearly if there is a strong physical change, then it is more likely that they will also be different biogeochemical regimes.*

*However, there can also be biogeochemical variability that is unrelated to physical variability, even variability in biogeochemical variables purely due to physical affection that is not apparent in the temperature and salinity fields. Thus the analysis presented here is closer to a necessary condition for a good Lagrangian trajectory, but is not necessarily a sufficient one.*

*The only true measure of biogeochemical variability around a quasi-Lagrangian trajectory is the actual variability. This will be different depending on the variable measured and the depth at which it is measured. Given the ability to make simultaneous measurements of many biogeochemical quantities from multiple drifting platforms, and the ability to autonomously survey around a platform, measuring in many places around a Lagrangian platform is becoming increasingly possible.*

Response:
The reviewer's main points are important and well-founded. Yes, the paper is focused on the data and context of OUTPACE, and other environments such as the North Atlantic Spring Bloom would have variability (perhaps both biogeochemical and physical) on much smaller scales. The hope in this manuscript is to generate a conversation regarding the best approach to ensure that biogeochemical measurements from drifting platforms are indeed reflective of a single biogeochemical environment's evolution, even when gradients exist at smaller scales.

As the reviewer points out, we use physical variability as a proxy for biogeochemical variability. The reviewer further adds that while a physical regime change can be indicative of a biogeochemical regime change, an absence of physical change does not necessarily preclude the existence of biogeochemical gradients. This is a critical point, and it represents a weakness in our approach. Indeed, as the reviewer suggests, our methodology may represent a necessary, but not sufficient, criteria to answer our question: Did the drifter and related station sampling stay in one biogeochemical context?

While we did include satellite data showing little detected surface gradients in SST and chl-a to support application of our method, this does not refute the Reviewer's point. To clarify this

caveat and bring it to the reader's attention, we will add the following changes to the manuscript:

> Introduction, Pg. 2, Line 26:
> … which is the focus of the present study.
> Before proceeding into this study's description of our methodology, a few remarks are needed regarding its applicability. We already mentioned that we will consider regions away from strong, organized mesoscale structure. Additionally, the method relies upon independent physical, not biogeochemical, measurements to indicate a change of water mass due to the drifter not being Lagrangian. This approach does not detect the existence of biogeochemical gradients in water that might exist on smaller scales, so future application of our method requires the user to apply contextual knowledge of their sampling region and keep this possibility in mind. For this study, a regional biogeochemical gradient was expected (Moutin et al., 2017) and rationales for this method's application will be provided.
> The Oligotrophy to UlTra-oligotrophy …

Reviewer:
*Specific comments Section 2.1 – How many drifters were deployed that were different from the drifting moorings? Where were they deployed relative to the moorings?*

Response: The SVP drifters were the only different drifters that were deployed, with their numbers provided in Table 1. The mean distance between their deployment position and the first SedTrap position will be added to the methods section:

> Sec. 2.1, Pg. 3, Line 14:
> Before starting each LD station, surface velocity program (SVP; Lumpkin and Pazos, 2007) drifters were deployed adjacent to the site. The numbers of drifters deployed are summarized in Table 1, and their grouped mean initial positions were 1.1, 1.6, and 0.9 km away from the start of station LDA, LDB, and LDC, respectively. At the start of each station, tTwo quasi-Lagrangian drifting moorings were deployed….

Reviewer:
*How were the SD stations placed relative to the LD stations? Was there any attempt to do a survey around the LD stations?*

Response: Since the LD stations were the focus of the manuscript, Fig. 1 did not include the SD station positions. They will be added to Fig. 1b in lieu of repeating the LD stations (see below). The SD stations were positioned to be roughly equidistant from each other, and site selection is further detailed in the preface of the special issue (Moutin et al., 2017).

[Figure]

Regarding attempts to survey around the LD stations, indeed there were surveys using a Moving Vessel Profiler (MVP), which produces high-resolution profiles. These data were used in another publication in this special issue focusing on LDB (de Verneil et al., 2017). However, as detailed in the supplementary information from that text, technical difficulties led to salinity variability that, while sufficiently accurate to recover profiles of the density stratification, were deemed to be too variable for direct use in calculating Spice and Z-score values. Generally speaking, having exploitable depth-resolved data at these intermediary scales of 10-100 km would indeed be very useful, and the collection of these data was included in our list of recommendations.

Reviewer:
*Figure S3 – Would be nice to additionally label the dispersion plot with units of meters and kilometers to guide the reader, i.e. 10^8 m^2 is (10 km)^2*

Response: A unit guide will be added to Fig. S3 (see below).

[Figure]

Reviewer:
*Recommendations – Although the first Rossby radius appears to be a good scale here, it is certainly much too large in other, more patchy environments.*

Response: We agree that in other environments that the Rossby radius may be too large a scale, and update the manuscript's recommendations as follows:

> Sec. 5, Pg. 17, Line 19:
> Upon arrival at the selected site, a deep CTD cast below the thermocline can be used to quickly calculate the local RD Rossby radius in real time and produce a rough estimate for maximum spatial scale. In patchier environments, this scale might be too large, and must be reduced.

---

## Author Comment (AC2) · 20 Feb 2018

Response to Review # 2 for OUTPACE LD Station Paper:

We thank Reviewer #2 for taking the time to read the submitted manuscript and providing their thoughts and comments. Below is the original response from Reviewer 2 (in italics), with our own responses interspersed within. Manuscript changes are shown with additions in blue, deletions in red strikethrough.

Reviewer:
*The paper assesses the Lagrangian nature of drifting sediment traps using physical data collected on a drifter and from ship observations of the surrounding waters. While the motivation of the study is sound I found the presentation difficult to follow. If the intention is to both assess the Lagrangian nature of the OUTPACE deployments and provide a generic method for such an assessment then the presentation of the method needs to be improved to enable the reader to clearly follow the approach. I consider the paper requires major revisions to address this point and hence be acceptable for publication. In revising the paper, the authors need to address:*

*1. The way the method section is written it is not clear that you are assessing the Lagrangian nature of the SedTrap drifter and how you use the collected data to do this. I like the use of Spice in the analysis but how is the baseline data profile determine – first hour, day? Of the SedTrap measurements? Something else? Need to clearly define the data that went into the baseline definition before I can assess the validity of the method. Sounds like you used the CTD data but I would think you should use the initial profile of the SedTrap drifter and then look at changes in the water properties of the SedTrap Drifter to determine whether the float is Lagrangian.*

Response:
        Due to the large number of datasets used in our analysis, we recognize that it can be difficult to follow the methods section. You read correctly, the baseline data is derived from the entire ensemble of the CTD rosette data, not the data from the SedTrap drifter itself. We understand that logically, the most direct evaluation of the Lagrangian strategy's ability to stay in one water mass is to analyze the timeseries data from the SedTrap drifter, and set a baseline from the start over a prescribed time period. First, the decision to specifically use the CTD rosette data instead of drifter data was made for a few reasons:
- CTD rosette data resolve a greater depth, and hence density, range and allow for comparison with the largest number of complementary datasets.
- While the station's sampling is defined by following the SedTrap drifter, the majority of the important measurements to achieve the goals of OUTPACE stem from water taken from the CTD rosette. Thus, while the drifter data are clearly important, the true ultimate need is to see whether the nearby CTD data also represent a single water mass.
- The sampling design for OUTPACE specifically included a large number of CTD casts, partly to provide the rich timeseries used in this analysis.

        For these reasons, we think the CTD data is still central to our analysis, and should remain the benchmark. The suggestion by the reviewer to conduct timeseries analysis is a good one. By taking entire datasets and comparing their differences over space, our method removed time-dependence within the datasets. In constructing the methodology of this manuscript, a few underlying assumptions were applied and perhaps not adequately stated, as pointed out in another Reviewer's comments. These hypotheses include:

- The SedTrap drifter, with multiple objects providing drag at different depths, will clearly not be Lagrangian for a long time.
- Vertical shear will be the norm, not the exception, for currents in the Ocean. Thus, the drifter will experience water advecting past it at different rates since the drifter is not truly Lagrangian.
- Field campaigns for these drifters will not be always associated with an evident physical structure, such as mesoscale eddies or fronts/filaments, that indicates a physical and/or biogeochemical gradient (perhaps by design, as in our case). Physical variability will then be residual gradients resulting from complicated stirring not readily discernable.

As a result of these hypotheses, the method was constructed to see what gradients existed and at what scales; having determined the scale, available data on currents was used to see if water from beyond this scale could plausibly have advected past the drifter. Conducting timeseries analysis on the drifter and CTD data is a pre-requisite before moving on to our methodology. Therefore, we have conducted this analysis.

The initial time period chosen for establishing the statistical baseline is the local inertial period (36.5, 38.3, and 38.0 hours for LDA, LDB, and LDC, respectively), so that internal wave effects are taken into account. Therefore, we have the assumption that over an inertial period the water mass has not changed in the CTD data. We will start by comparing the baseline to the Drifter timeseries. Our initial plots of the Z-score timeseries show that the drifter sensors, fixed at a given depth, experienced different densities due to internal waves (see below for an LDA example; black lines are Z=2 thresholds, magenta line indicates end-time for CTD baseline determination).

[Figure]

When these waves passed through, clear trends in Z-score were visible. This indicates a density-spice relationship present even at sub-bin scales. Therefore, the statistical baseline was redefined with a functional fit. This fit consisted of taking the ensemble of sorted density-spice pairs from the CTD baseline period, interpolating to a regular grid at 4 times the average density observation spacing, and smoothing with a moving average over the a bin width of +/- 0.1 kg m$^{-3}$. Standard deviations were calculated in a similar fashion, using observations within a moving window of the same bin-width. The new baseline distribution is shown below:

[Figure]

With this functional form, it is now possible to calculate Z-scores for the entire timeseries of the drifter data by looking up the corresponding functional spice value for observed density and comparing it to the observed spice. The resulting Z-score timeseries for the six drifting mooring sensors are shown for LDA below:

[Figure]

The sensor data generally show |Z|<2, though the sensor nearest the surface showed a departure with Z>2, though the Z-scores goes back down. For the third depth, (88 m), the data show some increasing variability with time though no trend is seen. The results for LDB are below:

[Figure]

For LDB, the surface similarly shows some |Z|>2 departures, but this is not permanent; the rest of the depths show variability around Z=0, with some observations at |Z|>2, though this is likewise not an irreversible trend. Larger variability is again found at 86-m depth.

[Figure]

The surface for LDC is similar to the other stations. At depth, however, LDC shows a different story. At 83 m, there are large departures and a slight positive trend, and at 105 m there is a clear trend and oscillations. Therefore, it is possible that different water advected towards the drifting mooring at these depths. Fitting a smoothed version of the timeseries (2-

hour moving window), and finding where they cross |Z|=2, provides an initial timestamp of concern at 3/27/2015 16:05:01 and 3/26/2015 17:10:00 GMT for the two depths.

Performing a similar timeseries analysis for the CTD data, we find similar patterns, as shown below. All observations are plotted in green, and all |Z|>2 plotted in black.

[Figure]

[Figure]

[Figure]

In these plots, we can see that LDA's Z-scores were close to zero, and that it is at the surface that we observe |Z|>2, similar to the SedTrap Drifter. During LDB, Z-scores were generally muted, with few observations surpassing |Z|>2. LDC shows a similar trend in increasing Z and variability as in the Drifters, and we can see that the variability is for densities between 1024 and 1024.5 kg m$^{-3}$, as well as a peak near 1025 kg m$^{-3}$.

We can also see that besides the time-dependence of the large Z-scores, that more and more density layers begin to be affected, starting halfway through March 27, 2015. Likewise, the peak near 1025 kg m$^{-3}$ begins around this time. Previously in our manuscript, we accepted this variability as part of the baseline's definition, and noticed that since the drifters had lower Z, that this would be acceptable. We thank the reviewer for their question that allows us to better understand the time-dependent nature of the variability.

These new findings require alterations to the Methods, Results, and Discussion. Since our responses to the reviewer's other points also require changes to these sections, the detailed amendments will be appended at the end. Here we provide a quick preview summary: Methods will be re-organized to describe the time-series as a first step in the analysis. Results will include the discussion above, with accompanying figures. The Discussion will include the implications of the gradients found during the drifter deployments.

Reviewer:

*2. The second step to the method is also not clear "evaluation of whether that scale was surpassed by the SedTrap Drifter ...". What does this mean, why is it useful, how is it determined, and why does it differ from looking at the changes in the water properties of the SedTrap Drifter? Please expand this discussion so I can follow how the current data collected on the different platforms are used to assess the Lagrangian nature of the SedTrap drifters. The trajectories from the different velocity datasets appear significantly different and not consistent with the drift of SedTraps, what does this mean?*

Response:

   As mentioned in our response to item 1, our assumption is that the drifter will not be Lagrangian for long periods. As a recourse, the original method's first step investigates at what spatial scale the water mass changes, and the second step uses complementary velocity data to see if water from these areas could have come into contact with the drifter. Logically, after conducting the timeseries analysis it may seem that if we do not observe changes in the timeseries, the analysis is finished. However, the physical changes determined by the baseline are relative and specific to the OUTPACE data. Therefore, in absolute terms, not observing a change in the timeseries by this metric does not mean that a change has not occurred.

   Since we are constrained to using the cruise data (i.e. baseline), one approach to better contextualize it is to analyze data from around and outside the deployment. If a strong physical change (e.g. Z-score) is seen outside of the sampling region, but is not observed in the CTD or drifter timeseries, then we are more assured that the water mass has not changed. If there is no change over large distances where one would expect at least some change, then this calls into question the timeseries results. Ultimately, this is why extending the Z-score analysis to greater spatial scales is used to create a spatial scale: since our baseline is relative, more data from different environments is needed to shore up the timeseries conclusions.

   Due to the failure of the drifter to be Lagrangian for long periods, and the presence of shear, it is plausible that certain density layers have advected from beyond the actual trajectory of the SedTrap drifter. This is why the velocity reconstruction is useful: if perceived changes in the timeseries are weak or marginal (so $|Z|<2$ or $|Z|\sim2$), then an additional check is to see if it is possible that sufficiently different water (as roughly determined by the spatial scale) could have reached the drifter. Therefore, this tests not so much if the drifter is Lagrangian, but rather that surrounding water masses did not impinge on the station's sampling in a region of apparently weak gradients.

   The fact that the trajectories are different from each other and the drifters reflects the general trend that a) velocities are intensified near the surface, and b) vertical shear at different layers is leading to different trajectories. The sum of the vertical shear acting upon the drifter might lead to less net displacement, though the water passing by indeed comes from farther afield.

   To clarify and better motivate also this second step, we have added the above discussion to the new version of the manuscript, in the Methods section. As before, these changes are reflected in the appended pages at the end.

Reviewer:
*3. The method section also provides much additional information that does not help clarify what you are doing. What is the value in comparing to climatology for the Lagrangian assessment? Why discuss water mass breakdown if it is not used? Why even show the remotely sensed maps and evolution if they are not used to assess the Lagrangian nature of the SedTrap Drifters?*

Response:

The climatology was added to show inherent variability in the region to inform whether the statistical baseline, which we have noted is relative, was reasonable. Having a second climatology, however, is perhaps not necessary, and so the second supplemental figure with the CARS data has been removed.

The water mass breakdown was mentioned to highlight the need for developing a new method in identifying the water mass seen during OUTPACE and how to establish whether it had changed during the SedTrap drifter's deployment. Since this is not necessary for the development of the method, we will remove mention of it in the Methods section.

The remotely sensed maps and evolution were included because a) as part of the OUTPACE special issue the spatio-temporal context of the three stations would be useful for readers (and authors) of the accompanying articles, and b) to demonstrate that no apparent gradients or changes (in SST or Chl-a) were visible around the SedTrap drifter. As another reviewer has pointed out, biogeochemical gradients do not necessarily coincide with physical ones. By including these data, we can show that no obvious variability was present for our stations, and suggest that the use of our method is valid.

Reviewer:
*4. Is the Lagrangian nature satisfied for all depths in the SedTrap drifter deployment? One may expected a surface drifter may not represent the flow at depth. Please discuss.*

Response:

The analysis conducted in response to item 1 shows that water mass changes could occur at some depths but not others (particularly at LDC!). Yes, a surface drifter may not represent flow at deeper depths, which is partly reflected by the trajectories of the SVP (bottom row of Fig. 8) against the SADCP trajectories (middle row of Fig. 8). As shown above, we have broken down our analysis to resolve depth-dependence, which will be reflected in the amended Results, Discussion, and Figures appended at the end of this document.

Reviewer:
*5. The SedTrap drifters all drift less than 5.6 km – hard to think it is not Lagrangian since it barely moved. Why do analysis out to 1000 km when the drifter barely moves? Need to provide some context as to why you extend the analysis to much larger scale.*

Response:

The SedTrap drifters did not drift very far, this is true. However, the trajectories calculated for different depths show that water at different depths probably did move much farther, and in different directions. The resultant drag force of these motions potentially cancel out, and so the fact the drifter did not move much does not preclude the drifter timeseries from observing water sourced from far away.

*Some specific comments.*
*Abstract. L1-5 – not clear what a quasi-Lagrangian drifter approach is – deploy a drifter and follow it. Add a sentence to explain what this is.*

Response:
We have added the following to the Abstract:
Pg1, Abstract, Line 2:

A popular experimental design is the quasi-Lagrangian drifter, often mounted with in situ incubations that follow the flow of water over time.  After initial drifter deployment, the ship tracks the drifter for continuing measurements that are supposed to  represent the same water environment.

*Pg3 line 23 – state how close the production line was to the SedTrap drifter?*

Response:
 Since the production line was recovered on a daily basis, it was followed closely by the vessel, and re-deployed close to the SedTrap drifter. While we do not have telemetry from the production line, the CTD positioning is the closest proxy. These distances range from 300 m to 5.7 km, averaging 1.3 km for the entire cruise, so we have added this to the manuscript:

Pg. 3, Sect. 2.1, Line 22:
 The Production Line was redployed on a daily basis close to the SedTrap Drifter. While no telemetry exists for the Production Line, the CTD casts from which incubation water was drawn ranged from 300 m to 5.7 km from the SedTrap drifter. After 5 days, …

*Pg4. How is the remote sensing data used to assess the Lagrangian nature of the SedTrap drifters?*
Response:
 As mentioned in our response to item 3, the remote sensing data were used to both justify the use of our method, and to provide context as part of the manuscript's role within the special issue of OUTPACE.

*- Where do you use Mixed Layer depth in your analysis?*

Response:
 The Mixed Layer depth was mentioned on Pg. 11, Sect. 3.3, Lines 10-11 to discuss how it was not resolved by the ADCP trajectories.

*Pg5 line8 – to make it easier to follow say you are assessing the SedTrap Drifters since it is confusing when you use the quasi Lagrangian drifting mooring.*

Response:
 This specific line has been removed as part of the larger changes to the Methods section, as shown above.

*Line 14 – how is comparing to climatology useful for your analysis? Line 18-25 – again interesting information but not relevant here. Could go in the introduction as a way of testing Lagrangian nature of the sampled water. It is not part of your method.*

Response:
 As previously discussed in the response to item 3, the climatology was used to motivate the statistical baseline. The section on the water mass breakdown will be removed (see above changes).

*Pg 6 line 14-25 – Z scores are functions of density do they vary significantly with density Line 34 – does the SedTrap Drifter have greater variability than TSG*

Response:

       The method has now been altered to retain differences in density, and are shown in the new figures presented in response to Item 1. For density-dependence in the baseline, we refer to Fig. 5, which has been updated and included in our response to Item 1. TSG variability, being near the surface, is generally greater than that of the SedTrap Drifter at depth.

*Pg 8 why is the satellite data needed?*

Response:

       As mentioned in our response to item 3, the remote sensing data were used to both justify the use of our method, and to provide context as part of the manuscript's role within the special issue of OUTPACE.

*Table 2 variation in Dist(z=2) is huge what does this mean? It implies the calculations are very dependent on the data source? Or you have not used the data appropriately*

Response:

       The variation in Dist(z=2) is due to our applying multiple functional fits. We did this because we did not assume a specific functional form for the Z-score vs. Distance relationship, and some variability is to expected. Also, as can be seen in Fig. 6, the TSG data is more variable, and the Z-scores increase much faster. Since surface water (in the mixed layer) is subject to stronger forcing than at depth, this makes the two datasets complementary but not necessarily comparable. However, since we are now considering all the density layers, fitting functions to each sub-set of data is somewhat intractable to present, and with the re-analysis conducted during the response to this review, it was more practical to choose a smallest scale were |Z|>2 between the TSG and CTD data. As a result, Table 2 has been removed, and the Methods and Results have been updated (See amendments above).

*Figure 1. state it is a weighting of the 42 days of data based on inverse distance squared from the ship track. What remotely sensed data is used and what is it resolution in time and space?*

Response:

       We have re-phrased the caption to highlight the weighting of the data corresponding to the cruise, and have added the data source:

       Figure 1. Weighted average satellite surface (a) chl-*a* and (b) SST for the 42 days of OUTPACE . data are weighted  based on normalized inverse distance squared  from the RV *L'Atalante*'s daily position . Shiptrack shown in white. LD station locations shown with black +'s. Domains used in Fig. 2 are shown by color-coded rectangles, with green for LDA, red for LDB, and blue for LDC. Chl-*a* and SST provided by CLS with support CNES.

*Figure 2. Is there weighting of the remotely sensed data? If you need to show remotely sensed data I would use figure 2 only.*

Response:

Figure 2's data is the same as Figure 1, and this will be added to the caption. The other reviewer wanted additional information that has been placed in Figure 1, so we will keep it.

*Figure 3. why show this plot? It is not a Lagrangian view of the data. It simply shows the variability in the region around the time of drifter deployment. How do you use it in your assessment?*

Response:
  As mentioned above, the satellite data were used to justify the use of our method and provide context for the cruise. For example, the heating at the end of LDA was reflected by the Z-scores near the surface, so having an independent timeseries showing this informs the Results and Discussion.

*Figure 4. Spice is not orthogonal to density – why not?*

Response:
  Spice is not orthogonal to density due to the aspect ratio of the plot. Properly adjusted, this becomes more apparent. Also, since density is a non-linear function of temperature and salinity, some of the inherent curvature makes it difficult to see the orthogonality. Please see the figure below showing a zoom-in and changing of the aspect ratio that demonstrates the orthogonality.

[Figure]

*Figure 6. How do you measure distance for SedTrap drifter? How is distance defined for the other data? Why go out to 1000 kms when the acceptable distance is less than 100 km?*

Response:
  Distance for the SedTrap drifter is determined from the closest CTD cast in time. The distance to the TSG, SD/LD stations are from the initial CTD cast location for the given LD station. However, since our analysis now just uses the SedTrap drifter for its timeseries, this

information for the SedTrap drifter is no longer necessary. Including data out to 1000 km is needed to help constrain the determination of $R_Z$. As visible in the LDA Z-score vs. distance plot of the CTD data, density layers at depth can show low Z-scores for 1000s of kms.

*Figure 7. Why are there such large differences between velocities in h, I, k, l panels?*

Reponse:
  The differences between the velocities are likely due to small-scale unresolved motions, as mentioned in Pg. 11, Sect. 3.3, Lines 18-19.

*Figure 8. rewrite caption to clarify what you are showing. What are the 3 rows showing? The trajectories appear quite different between the rows, why? And what implications does this have for the Lagrangian assessment?*

Response:
  The trajectories appear quite different between the rows because they represent either satellite-derived geostrophic flows (top row), ADCP currents throughout the water column (middle row), or SVP drifters representing the surface. The top row, showing only the geostrophic surface current, does not include waves. These waves, with depth-dependent structure, lead to the various trajectories seen in the ADCP-derived middle row. The bottom row's SVP trajectories represent the surface, but in the mixed layer not resolved by the ADCP. As a result, we feel it is not entirely surprising that these trajectories are so different, and so each contribute in their own way to the analysis. The implication is that satellite data are not necessarily sufficient for a Lagrangian assessment, and that different depths will travel different distances, so figuring out where the physical gradients are for each depth is important for the Lagrangian assessment.

The caption will be rewritten as follows:

[revised manuscript text omitted]
=2)$ () 96 437 46 -344 Exponential $r^2$ 0.79 0.87 0.83 -0.32 $Dist(Z=2)$ () 0.3 809 85 -7.4×10$^3$

 **LDB** Linear $r^2$ 0.78 0.60 0.90 0.18 $Dist(Z=2)$ () 21 658 37 89 Exponential $r^2$ 0.77 0.56 0.81 0.04 $Dist(Z=2)$ () -30 900 57 -990

**LDC** Linear $r^2$ 0.54 0.68 0.73 0.25 $Dist(Z=2)$ () 43 726 56 -355 Exponential $r^2$ 0.50 0.72 0.67 0.10 $Dist(Z=2)$ () -49 1.05×10$^3$ 51 -3×10$^3$

Supplementary Figure 2. T-S Diagrams like in Fig. S1, but with the CSIRO Atlast of Regional Seas (CARS) climatology.

[Figure]

**Figure 4.** T-S diagrams of (a) LDA, (b) LDB, and (c) LDC and surrounding stations. LD stations are color-coded, and SD stations different shades of gray. Isopycnals are displayed in black, with isopleths of spice shown in red.

[Figure]

**Figure 5.** Statistical LD baseline of spice versus potential density for (a) LDA, (b) LDC, and (c) LDC. Mean values (dots) plotted with intervals in between envelope of two standard error $\pm 2\ SErr_{obs}$.

[Figure]

**Figure 6.**  Z-score  timeseries for a) 14, b) 55, c) 88, (d) 105, (e) 137, and f) 197 m depth.  End of inertial period timeframe for  baseline definition plotted in magenta.  Z=-2 and 2  plotted  in black

[Figure]

**Figure 7.** Same as Fig. 6 but for LDB.

[Figure]

**Figure 8.** Same as Fig. 6 but for LDC.

[Figure]

**Figure 9.** Timeseries of CTD observations for LDA. Observations with |Z|<2 plotted in green, |Z|>2 in black. End of statistical baseline definition period shown in magenta.

[Figure]

**Figure 10.** Same as Fig. 9 for LDB, with |Z|<2 observations plotted in red.

[Figure]

**Figure 11.** Same as Fig. 9 for LDC, with |Z|<2 observations plotted in blue.

[Figure]

**Figure 12.** TSG Z-score over distance for LDA( left, green), LDB (center, red), and LDC (right, blue). |Z|>2 shaded black. Rossby radii $R_D$ distance plotted in horizontal dashed lines, color-coded to the LD stations.

[Figure]

**Figure 13.** SD station Z-score over distance for (a) LDA, (b) LDB, and (c) LDC. |Z|>2 shaded black. Rossby radii $R_D$ distance plotted in horizontal dashed lines, color-coded to the LD stations.

[Figure]

**Figure 15.**  Observed and  calculated trajectories for  currents analysis. Data from LDA,  LDB, and shown in left, icenter, and right columns, respectively. Top row (a-c): Observed trajectory of SedTrap drifter plotted in magenta. Time-averaged altimetry-derived surface currents shown with black arrows. Rossby radius RD traced as a color-coded circle, and RZ, the calculated spatial scale, in black. Starting position of SedTrap  drifter shown with a  star. Middle row (d-f): Calculated SADCP 38 kHz trajectories of water at each depth down to 600 m. Bottom row (g-i): Observed SVP drifter trajectories, with mean trajectory plotted  in black.

---

## Author Comment (AC3) · 20 Feb 2018

We thank Dr. Xavier Capet for his time and effort in both reading the submitted manuscript and providing his thoughts and comments. Below is his original post (in italics), with our own responses interspersed within. Manuscript additions are written in blue, with deletions in red strikethrough.

Reviewer:
*This manuscript is concerned with the evaluation of the Lagrangian nature of series of measurements made using a drifting mooring as a reference center. Specifically, the main goal of the study is to define an objective measure to quantify the degree to which sampling along the trajectory of a drifting mooring can be considered Lagrangian (with the ability to choose a threshold value for that measure beyond which the observations will not be considered Lagrangian). This is a methodological topic, yet it should be of interest to a relatively broad audience concerned with physical and biogeochemical ocean observations. The text, figures and captions are clear and informative (except in very few places, see below). On the other hand, the scientific context and working hypotheses underlying the study are not adequately posed in my opinion. Suggestions are provided below on how to improve this, which may simply involve modifications of the text. Alternatively, a more in-depth revision could make use of numerical simulations to carefully evaluate the proposed Lagrangian evaluation method.*

*Main comment: the scientific context and working hypotheses underlying the study are not adequately posed.*

*This is perhaps due to the fact that the authors are venturing into "new territory." There have been past studies attempting to quantify departures from pure Lagrangian motion associated with drifter trajectories (D'Asaro 2003) but the context of the present study differs. Indeed, the drifting mooring deployed during OUTPACE are not expected to precisely follow a water parcel of infinitesimal size but rather to remain in a given environment having a finite vertical height (defined in terms of density range in that study). The reader attention should be clearly drawn to this specificity at the beginning of the paper. An important consequence noticed by the authors (but in their final sections) is that, except in a perfectly barotropic flow, the mooring trajectory will not be truly Lagrangian at any vertical level. In this context, and ignoring vertical motion, the three key elements of the problem are 1) the vertical shear of the flow 2) the structure of the mooring and the vertical distribution of the drag force 3) the horizontal scales over which the environment can be considered homogeneous (the scale may vary as a function of depth, e.g. scales may be shorter near the surface where submesoscale turbulence tends to be intensified).*

*Several remarks are in order. With respect to 3) it seems difficult, except in rare cases, to define an average scale as done in the present study. Indeed the Rz defined by the authors ignores the Lagrangian aspect of the circulation. A fast moving environment eg in the vicinity of a hyperbolic point, will be systematically poorly evaluated from the perspective of the ability of OUTPACE moorings to remain Lagrangian, but for erroneous reasons because only the intensity of the shear (which may or may not be stronger in such regions) matters. Likewise, a mooring embedded into a mesoscale structure may lead to a near-perfect "Lagrangian trajectory" (in the sense that the mooring perfectly tracks at all depths the coherent water masses trapped into the eddy) while traveling distances greater than Rd (eg, on time scales of weeks to months). I understand that the OUTPACE setting is one in which mesoscale are weak and this is explicitly stated by the authors. But then, could the authors strive to precisely describe the kind of dynamical regime they are in, and how it affects the*

*general problem that is the motivation of the study? If mesoscale turbulence is irrelevant then waves with different scales should dominate, typically Rossby waves, inertial waves, and tides. And only the motions with time scales longer than the station duration impact the "quasi-Lagrangian strategy" through their vertical shear. This is explicitly mentioned by the authors (p15 lines 23-25) but again, these concluding remarks come way too late and should have guided the whole validation design.*

*In any event, whether or not the presented work is only valid in one regime or not, it is unlikely that a proper metric for how Lagrangian the observations are can exclude the measured vertical shear in the area. As for the rule of thumb that displacements longer than Rd should raise suspicion, I would certainly argue that this is too general a statement to be supported by the present work. Likewise the suggestion that there might be inappropriate flow regimes (p17 line 15) is not well supported and may have to be reconsidered.*

*Overall, it seems to me that the introduction should i) present the processes that can break the Lagrangian nature of the sampling approach implemented by the authors ii) briefly review how different dynamical regimes may be differently affected 3) present the OUTPACE regime of interest in which they will develop their methodological approach. This would allow the authors to put their work on much firmer ground and help them write a more robust discussion section.*

Response: We agree that the concerns brought up in these comments should be addressed early in the manuscript, particularly in the introduction. The difference between neutrally buoyant drifters such as in D'Asaro (2003) and quasi-Lagrangian platforms, and the expectation that the latter will not be particularly Lagrangian for an extended period, needs to be highlighted. Also, the flow regimes that complicate these drifter deployments (and not treated by this method) will be specified. Finally, a characterization of the regime we are targeting is included:

> Sect. 1, Pg. 2, Line 16: Naturally, the question arises whether the trajectory undertaken by the drifting mooring in the quasi-Lagrangian approach accurately represents the water movements at each of the sampling sites. If the drifter is successful in following the water, then indeed a single biogeochemical setting will have been sampled; if not successful, then the risk grows that a different environment has been brought in via advection. Previous efforts by physicists to make floats Lagrangian show the effort needed to make an instrument neutrally buoyant, and they have been instrumental in demonstrating complicated flow regimes (D'Asaro et al., 2011). In contrast, the quasi-Lagrangian platform, with a variable distribution of incubation bottles, will necessarily fail to be Lagrangian in finite time outside of a barotropic flow where currents are the same throughout the water column. Ensuring the success of this strategy thus requires taking into account how different currents potentially shorten the timespan of validity. In fast-moving flows with strong vertical shear and possible vertical motions, such as zones of enhanced mesoscale turbulence near fronts and filaments, the drifter will not be Lagrangian long, perhaps not even a day. Alternatively, if a drifter is trapped inside a coherent eddy, it can follow a similar water mass for a long time (weeks to even months) over great distances. Periodic vertical shear can also arise due to waves and tides, so displacements due to them must also be considered. In deployments lasting several days but less than a week, these are longer than internal waves, inertial frequencies and tides which we will consider, but less than the

passage of Rossby waves which will be ignored. As a result of all these possible motions, it is therefore necessary to carefully consider site selection and dynamical regime before deployment. Unless the focus of study, fronts and filaments might be avoided because constant shearing will quickly separate water parcels at different depths in the direction of the structure's alignment. Fortunately, finding signs of their presence has become more feasible with satellites. An eddy can be targeted because of its coherence, and there are ways to confirm that sampling is indeed inside of it (Moutin and Prieur, 2012). In other words, if a physical structure is targeted or identified, its particular nature supercedes other considerations. These structures are not necessarily representative of the world Ocean, and so for biogeochemical measurements to reflect predominant conditions it is necessary to sample elsewhere. For the campaigns where sites are far (possibly by design) from obvious, organized mesoscale structure and lasting around several days to a week, there is still a need to conduct an independent, post-cruise validation of the drifter's success, ~~In practical terms, ensuring the success of the quasi-Lagrangian strategy leads to a two-step process. First, the selection of sampling sites needs to be carefully considered. Sites should be relatively uniform to provide room for error should the drifting mooring wander too far from the true water displacement. Prior to the advent of satellite oceanography, structures harboring enhanced gradients such as eddies and fronts were difficult to detect before sampling. As a result, the risk always existed that sites could be chosen close to these structures, putting the drifter's mission into jeopardy. In recent years, the incorporation of near-real-time satellite data having become routine minimizes this (in the list above, since the BOUM 2008 campaign; Moutin et al., 2012). The second step, after deployment of the drifting mooring, consists of an independent, post-cruise validation of the data~~, which is the focus of the present study.

Reviewer:
*If the authors intend to keep using this type of observational approach, I would additionally suggest a numerical investigation in which drifter trajectories advected with velocities computed by several weighted vertical averages are compared with trajectories for drifters that are localized at particular depths and thus Lagrangian (if w can be ignored). Such a study might well reveal the broad range of $R_z$ spatial scales coexisting in a given location even with limited mesoscale activity. A numerical investigation with biogeochemistry would even allow the authors to explore the limitations raised by reviewer 1.*

Response: Conducting a numerical study, possibly with a biogeochemical component, would indeed be a necessary step to refine this methodology for the various flow regime scenarios brought up in these remarks. We feel, however, it is beyond the scope of this paper. As part of the OUTPACE special issue, this paper's aims are to provide context for these specific deployed drifter samples and evaluate their validity in representing individual environments, albeit embedded within flow regimes that we feel are appropriate for our chosen methodology. Seeing as how the quasi-Lagrangian platform is a popular one, and will continue to be used by numerous research groups, we welcome the suggestion and invite a potential future collaboration to expressly explore its limitations.

Reviewer:
*Minor comments: abstract: "homogeneous" may be better suited than "self-similar".*
*P4 line 10: "Square regions…" I find this sentence unclear. P9 line 22: Not clear. Rephrase perhaps as: "The increase in salinity maximum from LDA to LDC reflects…"*

*P15 line 33: "Rather than an elevated shearing apart…". Awkward sentence*

*Reference: D'Asaro, E.A. (2003). Performance of autonomous Lagrangian floats. Journal of Atmospheric and Oceanic Technology, 20(6), 896-911.*

Reponse: The following changes will be made to the manuscript:

Sec. Abstract, Pg. 1, Line 10:
…their own sufficiently homogeneous  physical environment …

Sec. 2.2, Pg. 4, Line 10:
 The spatial range consisted of a 120 x 120 km box centered at each LD station.

Sec. 3.2, Pg. 9, Line 22:
The increase in salinity maximum from LDA to LDC  reflects  the high salinity tongue of the South Pacific…

Sec. 4.2, Pg. 15, Line 33:
However, the  exponential growth rate was so low that it took over a week for the relative …

---

## Author Response (AR1)

Dear Associate Editor L. Memery and to whom it may concern,

My co-authors and I would like to first convey our thanks for your time and ongoing effort serving as associate editor for our manuscript, "OUTPACE long duration stations: physical variability, context of biogeochemical sampling, and evaluation of sampling strategy." In response to your comments and suggestions regarding the first revisions of our manuscript, we have made further changes to the Abstract, Results, Discussion, and Conclusions sections, as well as an updated version of Fig. 15, as requested. The figures added in the previous revision have been refined, as well.

The marked-up pages showing the changes made since the first revision immediately follow this cover letter, and we hope they satisfactorily address your concerns. Additionally, since the changes made from the initial submission were not centralized into a single document, but rather distributed between the three responses to the reviewers, a marked-up version of the entire manuscript, reflecting all changes made, has been appended.

We thank you in advance for considering this revised manuscript for publication in Biogeosciences.

Sincerely,
Alain de Verneil

[revised manuscript text omitted]